# Effect of Chemical Treatment of Cotton Stalk Fibers on the Mechanical and Thermal Properties of PLA/PP Blended Composites

**DOI:** 10.3390/polym16121641

**Published:** 2024-06-10

**Authors:** Feng Xu, Jin Shang, Abdukeyum Abdurexit, Ruxangul Jamal, Tursun Abdiryim, Zhiwei Li, Jiangan You, Jin Wei, Erman Su, Longjiang Huang

**Affiliations:** 1State Key Laboratory of Chemistry and Utilization of Carbon Based Energy Resources, College of Chemistry, Xinjiang University, Urumqi 830017, China; feng.xu@xju.edu.cn (F.X.); shangjinpop@163.com (J.S.); jayou@xju.edu.cn (J.Y.); 17793738494@163.com (J.W.); 2State Key Laboratory of Chemistry and Utilization of Carbon Based Energy Resources, State Key Laboratory of Oil and Gas Fine Chemicals, Ministry of Education & Xinjiang Uygur Autonomous Region, Xinjiang University, Urumqi 830017, China; abdukaiyum@sohu.com (A.A.); li2812355161@163.com (Z.L.); suerman47@163.com (E.S.); 18034152696@163.com (L.H.)

**Keywords:** polylactic acid, cotton stalk fibers, natural fiber reinforced composites, chemical treatment

## Abstract

Different chemical treatment methods were employed to modify the surface of cotton stalk fibers, which were then utilized as fillers in composite materials. These treated fibers were incorporated into polylactic acid/polypropylene melt blends using the melt blending technique. Results indicated that increasing the surface roughness of cotton stalk fibers could enhance the overall mechanical properties of the composite materials, albeit potentially leading to poor fiber–matrix compatibility. Conversely, a smooth fiber surface was found to improve compatibility with polylactic acid, while Si-O-C silane coating increased fiber regularity and interfacial interaction with the matrix, thereby enhancing heat resistance. The mechanical properties and thermal stability of the composite materials made from alkali/silane-treated fibers exhibited the most significant improvement. Furthermore, better dispersion of fibers in the matrix and more regular fiber orientation were conducive to increasing the overall crystallinity of the composite materials. However, such fiber distribution was not favorable for enhancing impact resistance, although this drawback could be mitigated by increasing the surface roughness of the reinforcing fibers.

## 1. Introduction

In recent years, there has been a growing interest in biodegradable and renewable polymers because of environmental concerns and sustainability issues associated with petroleum-based polymers [1,2,3,4]. Within the realm of biodegradable polymers, polylactic acid (PLA) has emerged as a leading contender in the field of biodegradable composites owing to its renewable nature, making it an excellent substitute for petrochemical-based polymers [5,6,7,8]. However, the inherent brittleness, limited thermal stability during processing, and low crystallinity have long posed significant challenges that impede its widespread application across various domains [5,9,10,11,12].

The utilization of natural fibers as reinforcement materials within PLA matrices has been a prominent subject of research [13,14]. Researchers have devoted considerable efforts to developing and evaluating polymer composites reinforced with various plant fibers. Examples include flax [15,16], jute [17,18], corn stalks [19], eucalyptus fibers [20], coconut shell fibers [21,22], and cotton stalk fibers [23]. Among these, cotton stalk fibers (CSFs), derived from the agricultural waste of the globally significant economic crop cotton, hold immense potential for development and utilization [23,24,25]. From a chemical composition perspective, CSFs, as lignocellulosic biomass, primarily consist of cellulose, hemicellulose, and lignin [25]. The presence of numerous hydroxyl groups within these constituents makes the fibers susceptible to chemical reactions. By chemically modifying these functional groups, the activation and introduction of new chemical moieties can occur. Chemical processing involves immersing fiber straw in water-soluble chemical solutions, such as potassium hydroxide, sodium hydroxide, and sulfuric acid, to selectively eliminate non-cellulosic components while retaining the functional cellulose constituents. Coupling agent treatment, waterproofing chemical modification, and heat treatment are employed to modify the morphology, roughness, and surface polarity of the fibers [26,27]. This enhances compatibility and interactions between CSFs and polymer matrices, ultimately leading to improved performance of resulting composite materials.

The study conducted by Thapliyal et al. [27] revealed that the utilization of waste natural fibers in the production of fiber composites not only addresses environmental concerns but also offers a viable approach to achieving circular economy objectives. Chaishome et al. [28] treated flax fibers with alkali and utilized them as fillers to prepare composite materials, resulting in enhanced thermal stability of the corresponding composites. Du et al. [29] conducted silane coupling treatment on pulp fibers, followed by melt blending with PLA, to fabricate silane-coupled pulp fiber-reinforced PLA composite materials. Their research revealed that the silane coupling-treated pulp fibers achieved reactive compatibility with PLA, resulting in improved mechanical properties compared with the untreated counterparts. Goriparthi et al. [30] utilized 5% NaOH, potassium permanganate acetone, benzoyl peroxide, and silane solutions for the surface treatment of jute fibers, followed by the fabrication of jute fiber/PLA composite materials. Their investigation revealed that the surface-treated composites exhibited higher abrasion resistance compared with the untreated counterparts. Among various chemical reagents, PLA/silane-treated jute composite materials demonstrated the optimal abrasion resistance, whereas PLA/alkali-treated jute composite materials exhibited the lowest abrasion resistance. Surface modification of fibers can indeed enhance various properties of composites. However, there is relatively limited research on the relationship between the chemical composition, surface functional group distribution of fibers after modification, and the various properties of composite materials. Additionally, there has been scant investigation into the regularities governing the relationship between the internal component morphology of fibers and the respective mechanisms affecting material properties.

In this study, a series of chemical treatments, including stearic acid treatment, alkali treatment, silane treatment, and alkali/silane treatment, were employed to modify the surface of CSFs, resulting in different chemically treated CSFs. Combining with a previous report [23], using polypropylene grafted with maleic anhydride (PP-g-MAH) as a compatibilizer, poly(lactic acid) (PLA)/polypropylene (PP) blended composites were prepared through melt extrusion method using different chemically treated CSFs, with a material ratio of PLA: 70 wt%, PP: 5 wt%, PP-g-MAH: 5 wt%, and CSF: 5 wt%. The mechanical properties, Vicat softening point (VST), thermal stability, crystallinity, etc., of the composites were investigated and discussed through mechanical performance testing, Vicat softening point testing, thermogravimetric analysis, differential scanning calorimetry, dynamic mechanical analysis, and microscopic morphology observation. Emphasis was placed on the discussion and analysis of the relationship between the chemical composition, surface functional group distribution of CSFs, and various properties of the composite materials, elucidating the respective mechanisms of the internal component morphology of CSFs on the properties of the materials. By summarizing and analyzing the regularities between the changes in material properties and structural composition, some contributions were made to elucidate the mechanisms and application methods of other plant fiber-reinforced polymers.

## 2. Materials and Methods

### 2.1. Materials

The PLA (grade 4032D) [MFR = 7 g/10 min (210 °C/2.16 kg)], supplied by Nature Works LLC. (Minnetonka, MN, USA), and PP (grade T30S) [MFR = 1.14 g/10 min (190 °C/2.16 Kg)], provided by Dushanzi Petrochemical Company (Kelamayi, China), were utilized for this study. Cotton stalks were procured from the local market [Density 1.21 g/cm^3^, Diameter 12–35 μm, Length 15–56 mm, Tensile Strength 287–597 MPa, Elongation at Break 2–10%]. Prior to experimentation, the sample underwent an air-drying process, followed by grinding and second sieving to obtain fractions with 60 mesh particle size. The compatibilizer, PP-g-MAH, was sourced from Dongguan Hongxing New Materials Company (Dongguan, China).

### 2.2. Chemical Modification of Cotton Stalk Fibers

#### 2.2.1. Stearic Acid Treatment

CSFs were immersed in a 1 w% solution of stearic acid alcohol at room temperature for 2 h, followed by filtration, removing residual stearic acid from the surface of the fibers using deionized water, and drying treatment. These fibers shall be referred to as CSF-a.

#### 2.2.2. Alkaline Treatment

CSFs were soaked in a 0.5 w% NaOH aqueous solution and continuously agitated at room temperature for 24 h. Subsequently, the resulting product was washed with deionized water to achieve neutrality, and finally, the product was dried and designated as CSF-b.

#### 2.2.3. Silane Treatment

CSFs were immersed in a 1 w% solution of γ-aminopropyltriethoxysilane (silane coupling agent KH-550); the solution was prepared by blending isopropanol and water at a ratio of 4:6, followed by agitation for 2 h and subsequent filtration. Subsequently, the treated CSFs were dried and designated as CSF-c.

#### 2.2.4. Alkali/Silane Treatment

The alkali-treated CSFs were immersed in a 1 w% solution of γ-aminopropyltriethoxysilane (silane coupling agent KH-550), which was prepared by mixing isopropanol and water at a ratio of 4:6. After stirring for 30 min, the fibers were further soaked for an additional 2 h. Subsequently, the solution was filtered, and the treated CSFs were dried, resulting in their designation as CSF-d.

After undergoing chemical treatment, all samples of CSFs were subjected to a drying process in an oven at 80 °C for 24 h to eliminate any remaining moisture. Subsequently, the fibers were further dried in a vacuum drying oven for 4 h to facilitate the preparation of the composites. The quantitative analysis of the chemical composition of CSFs with various chemical treatments was conducted using the Van Soest method [31].

### 2.3. Preparation of Composite Materials

The raw materials are weighed based on the mass ratios provided in Table 1 and thoroughly mixed using a V-type high-efficiency mixer. Subsequently, the mixtures with varying proportions are subjected to an 80 °C hot air convection oven for 4 h to eliminate moisture from the samples. The dried materials are then processed through a twin-screw extruder for melt blending, subsequently pelletized using a pelletizer to obtain polymer materials with diverse chemical compositions. The pellets undergo drying in an 80 °C hot air convection oven for 8 h, after which standard test specimens are prepared utilizing an injection molding machine.

### 2.4. Characterization Studies

#### 2.4.1. Infrared Spectroscopy Analysis

The sample was subjected to chemical analysis using a Fourier-transform infrared spectrometer (FTIR) spectrometer (Thermo Scientific Nicolet IS20, Waltham, MA, USA). The spectra were recorded by performing 32 scans over the wavenumber range of 400–4000 cm^−1^.

#### 2.4.2. Heat Resistance

The Vicat softening temperature (VST) of the composite materials was evaluated using an HDT-Vicat tester in accordance with ISO 306:2013 standards [32].

#### 2.4.3. Mechanical Properties

The evaluation of the mechanical characteristics of both PLA and composites was carried out using a Universal Testing Machine sourced from Shenzhen WANCE Testing Equipment Co., Ltd., Shenzhen, China. Tensile tests were performed following the ISO 527-2 standard [33]. The impact strength was evaluated by employing notched specimens derived from the composites, and the assessments were conducted using an impact testing apparatus. Five parallel samples were selected for each test to ensure the representativeness of these data. Subsequently, the mean values obtained from these repetitions were documented along with their respective standard deviations.

#### 2.4.4. Morphological Studies

Investigate the microstructure of composite materials utilizing a Zeiss Sigma 300 scanning electron microscope (SEM) manufactured by Zeiss, Oberkochen, Germany. Before conducting SEM analysis, a fine layer of gold was deposited onto the sample surfaces using a sputtering device to enhance conductivity and reduce charging phenomena. Thus capturing high-resolution insights into the surface morphology of the samples.

#### 2.4.5. Differential Scanning Calorimetry

The melting and crystallization characteristics of the specimens were investigated utilizing the TA Instruments’ DSC250 Differential Scanning Calorimeter (DSC), sourced from Newburyport, MA, USA. Within the temperature range of −25 °C to 180 °C, the specimens were subjected to thermal cycling in a nitrogen gas environment. Calculate the crystallization rate utilizing the subsequent formula:$$X_{c}(\%)=\frac{\Delta H_{m}-\Delta H_{c}}{W_{PP}\times \Delta H_{PP}+W_{PLA}\times \Delta H_{PLA}}\times 100$$
where $\Delta H_{m}$ and $\Delta H_{c}$ represent the respective enthalpies of melting and crystallization for the specimens. where $\Delta H_{PP}$ and $\Delta H_{PLA}$, respectively, denote the enthalpy change upon complete crystallization of *PLA* and *PP*, measured as 93.6 J/g and 188.9 J/g.

#### 2.4.6. Thermal Properties

To assess the thermal stability of the composites, thermogravimetric analysis (TGA) was executed utilizing a Netzsch TG 209 F3 Tarsus thermogravimetric analyzer sourced from Netzsch GmbH, Selb, Germany. A 5 mg sample was placed into the instrument, and the heating temperature range was set from 30 °C to 600 °C.

#### 2.4.7. Dynamic Mechanical Analysis

The dynamic mechanical properties of the composite materials were examined utilizing a PerkinElmer 8000 Dynamic Mechanical Analysis (DMA) instrument, sourced from PerkinElmer Inc., Waltham, MA, USA, and equipped with a single cantilever fixture.

#### 2.4.8. Wettability Analysis

The determination of the samples’ hydrophilic and hydrophobic characteristics was conducted utilizing a German Dataphysics OCA25 contact angle measuring instrument, sourced from Dataphysics Instruments GmbH, Filderstadt, Germany. The specimens were repositioned onto a leveled platform, and surface analysis was performed through a water droplet examination. The droplet deposition process was recorded. The contact angle data of the sample surface were calculated using the SCA20 (dpiMAX 20P1) software.

#### 2.4.9. Moisture Absorption Analysis

According to the GB/T 1034-2008 standard [34] for testing material moisture absorption, five parallel samples were selected for each test to ensure the representativeness of these data. Subsequently, the mean values obtained from these repetitions were documented along with their respective standard deviations.

#### 2.4.10. Statistical Analysis

The basic parameters were visualized using Origin 2023b software (OriginLab Corporation, Northampton, MA, USA). Statistical analysis was conducted using the SPSS 25.0 software package (IBM Corp., Armonk, NY, USA). Data were presented as mean ± standard deviation (SD), and differences in experimental data were assessed by one-way analysis of variance (ANOVA) to determine statistical significance. A *p*-value less than 0.05 was considered statistically significant.

## 3. Results and Discussion

### 3.1. Infrared Spectroscopic Analysis of Cotton Stalk Fibers after Different Chemical Treatments

CSFs are primarily composed of three natural polymers: cellulose, hemicellulose, and lignin [35]. The surface structure of CSFs following different chemical treatments was characterized using Fourier transform infrared spectroscopy (FTIR), as illustrated in Figure 1. Combining the results of the compositional analysis of cotton stalk fibers under different treatments, as shown in Table 2 a comprehensive discussion of the effects of various chemical treatments on cotton stalk fibers is presented.

In the infrared spectra of CSFs, a prominent peak at 3318.98 cm^−1^ corresponding to the stretching vibration of hydroxyl groups (-OH) was observed. Additionally, a peak indicative of the presence of carbon–hydrogen bonds (C-H) was observed at 2921.61 cm^−1^. The peak detected at 1054.22 cm^−1^ corresponds to the vibration of C-O-C in cellulose and hemicellulose, indicating the presence of polysaccharide chains in CSFs [36]. Furthermore, the presence of benzene ring structures in CSFs led to characteristic peaks associated with aromatic rings in the range of 1241 to 1603 cm^−1^ [37]. The infrared peak corresponding to the hydroxyl groups (3318.98 cm^−1^) of CSFs treated with stearic acid (CSF-a) showed a reduction. This reduction is likely due to a chemical reaction between stearic acid and the hydroxyl groups on the CSFs, resulting in a decrease in the hydroxyl content on the fiber surface and the formation of ester bonds, thereby enhancing the infrared peak representing carbon–oxygen bonds (C=O) at 1748 cm^−1^. Additionally, the presence of stearic acid’s eighteen carbon alkane chain resulted in the encapsulation of the fiber surface, weakening the infrared peak associated with hydroxyl groups. The infrared spectrum of CSFs treated with alkali (CSF-b) reveals an enhancement in the infrared peak attributable to the stretching vibration of carbon–hydrogen bonds (2921.61 cm^−1^) and the hydroxyl groups represented at 3318.98 cm^−1^. This enhancement is attributable to the alkali treatment’s ability to remove pectin, polysaccharides, and other components from the surface of CSFs. Additionally, a decrease in the intensity of characteristic peaks related to aromatic groups observed between 1241 and 1603 cm^−1^, is noted. This decrease is attributable to the removal of lignin components by the alkali treatment, resulting in surface roughening and the exposure of a large number of cellulose carbon–hydrogen bonds and hydroxyl groups [38,39,40,41,42]. The treatment of CSFs with a silane coupling agent (KH550) (CSF-c) leads to a reduction in the intensity of the hydroxyl group peak at 3318.98 cm^−1^, attributable to the effect of the silane chain. Additionally, an enhancement in the peaks corresponding to the Si-O-C functional group is observed at 1054 cm^−1^ and 1105 cm^−1^. In comparison to CSFs treated solely with silane, those subjected to a combined treatment of alkali and silane (CSF-d) exhibit a strengthened hydroxyl peak at 3318.98 cm^−1^, albeit lower than that of alkali-treated fibers. Furthermore, a decrease in the peaks representing specific functional groups of lignin is noticed in the range of 1241 to 1603 cm^−1^. This phenomenon arises from the removal of lignin and pectin components due to alkali treatment, exposing a considerable number of hydroxyl groups. Subsequent silane treatment results in a reaction with these exposed hydroxyl groups, leading to a reduction in their peak intensity [43,44,45].

### 3.2. Microscopic Morphology of Cotton Stalk Fibers after Different Chemical Treatments

The reaction mechanism and morphological changes in CSFs with different chemical treatments are depicted in Figure 2. Scanning electron microscopy (SEM) was employed to conduct a microscopic morphological analysis of the treated CSFs, as illustrated in Figure 3. The untreated CSFs depicted in Figure 3a exhibit irregularities and gaps on the fiber surface, which can be attributed to the predominant cellulose-based fiber bundles forming the primary matrix of these fibers. These fiber bundles are held together by interfaces composed of pectin, hemicellulose, and lignin. Consequently, the presence of waxy layers and protrusions on the surface is clearly evident [46,47,48]. In Figure 3b, the image depicts CSFs that have been treated with stearic acid, resulting in a stearic acid coating on the surface of the fibers. This coating imparts smoothness to the surface and eliminates any impurities or protrusions present on the original surface of the CSFs. Figure 3c illustrates CSFs that have undergone alkali treatment, leading to the removal of components such as pectin, hemicellulose, and lignin from the fiber surface. Consequently, this process results in an increase in the roughness of the fiber surface, rendering it irregular and porous. Such a rough and porous surface facilitates enhanced adhesion between fibers and matrix materials by promoting mechanical interlocking between them [38,46,47,49]. The surface of CSFs treated with silane is depicted in Figure 3d. It can be observed that the application of a silane coupling agent induces significant alterations in both morphology and texture, resulting in a cleaner surface covered by a layer of silane coating. Additionally, the fiber surface exhibits uniformly distributed pores. This phenomenon can be attributed to the infiltration of the silane coupling agent into gaps on the cotton stalk fiber surface within an isopropanol/water mixture solvent, leading to the dissolution of impurities such as pectin under the influence of isopropanol. Subsequently, a hydrolysis reaction occurs when the silane coupling agent comes into contact with water, causing trimethoxy groups to hydrolyze into hydroxyl groups and form γ-aminopropyl dimethoxysilanol. Through condensation reactions between hydroxyl groups present on both the silane coupling agent and cotton stalk fiber surface, silicon–oxygen bonds and carbon–oxygen bonds are formed, ultimately resulting in the formation of a distinct silane coating on the cotton stalk’s surface while simultaneously altering its morphology [44,49]. The surface morphology of CSFs subjected to alkali/silane treatment is depicted in Figure 3(e1,e2). Initially, the fibers undergo alkali treatment, resulting in the removal of components such as pectin, hemicellulose, and lignin from the fiber surface. This leads to an increase in surface roughness and a proliferation of hydroxyl groups. Consequently, the fiber surface becomes irregular and porous, with numerous exposed hydroxyl groups. Subsequent silane treatment on the alkali-treated fibers results in the deposition of a silane coating on the rough and hydroxyl-rich fiber surface. This process yields a distinctive bamboo-like pattern on the fiber surface. As a result, compared with alkali-treated CSFs, these fibers exhibit smoother surfaces with reduced and evenly distributed voids.

### 3.3. FTIR Analysis of Composites

The characterization of the structure of composite materials derived from differently treated CSFs was performed using Fourier-transform infrared spectroscopy (FTIR). Detailed infrared spectra are depicted in Figure 4. The infrared spectra of all chemically treated cotton stalk fiber-reinforced PLA composites closely resemble those observed in untreated cotton stalk fiber-reinforced composites [40,44]. The infrared spectrum displays characteristic peaks corresponding to specific vibrational modes, including the stretching vibration peak of C-H at 2996 cm^−1^, the stretching vibration peak of C=O at 1748 cm^−1^, the bending vibration peak of C-H at 1450 cm^−1^, the stretching vibration peaks of O-C-O at 1181 cm^−1^ and 1081 cm^−1^, and the stretching vibration peak of C-C at 869 cm^−1^ [50]. Additionally, distinct absorption peaks corresponding to the stretching vibrations of aliphatic C-H bonds inherent to olefinic compounds are observed at 2950 cm^−1^, 2925 cm^−1^, and 2850 cm^−1^ [51]. These observed peaks align with previously reported infrared findings in the literature [51,52,53]. No additional peaks were observed in the infrared spectrum of the composite material derived from chemically treated CSFs, potentially attributable to the congruity between the newly introduced chemical bonds within the material and the inherent functional group structure of PLA itself, leading to overlapping characteristic peaks akin to those detected in PLA [54]. The intensity of the stretching vibration peak at 1748 cm^−1^ for C=O is correlated with the chemical treatment methods applied to CSFs, ranked in order from strongest to weakest as follows: N-CSF > NS-CSF > S-CSF > SA-CSF > UN-CSF > PLA [40]. This correlation can be attributed to the interaction between CSFs and PLA, which primarily occurs through bonding between hydroxyl groups (-OH) in CSFs and carbonyl groups (C=O) and carboxyl groups (-COOH) in PLA. Due to the presence of pectin and wax materials on the surface of untreated CSFs, a limited number of -OH groups are exposed, which can interact with the carbonyl (C=O) and carboxyl (-COOH) groups of PLA through hydrogen bonding and covalent bonding. In contrast, alkali-treated fibers (N-CSF) remove pectin and wax materials from the surface, increasing the available number of OH groups. Therefore, increased exposure of -OH groups in the fibers enhances their interaction points with the PLA carbonyl (C=O) and carboxyl (-COOH) groups through hydrogen bonding and covalent bonding. The silanized cotton stalk fiber surface possesses various functional groups, such as -OH and =NH, which can form hydrogen bonds and covalent bonds, respectively, with the PLA carbonyl (C=O) and carboxyl (-COOH) groups. In SA-CSF, a layer of stearic acid coating on the surface results in stretching vibration peaks caused by its own C=O functional group being higher than that of untreated UN-CSF.

### 3.4. Analysis of the Mechanical Properties of Composites

Different composite materials prepared from CSFs treated with various chemical processes exhibit distinct changes in their mechanical properties. It is crucial to analyze the performance of these composite materials for their diverse applications in real-life scenarios. Figure 5 presents a comprehensive comparison of the mechanical properties of these composites. Specifically, Figure 5a illustrates stress–strain curves obtained from tensile tests, Figure 5b compares the tensile strength and fracture elongation of each material, Figure 5c displays the flexural strength obtained from bending tests, and Figure 5d shows the impact strength comparison using a polymer V-notch.

The tensile stress–strain curves of the composite materials in Figure 5a clearly demonstrate that different chemical treatments have resulted in varying degrees of alteration in the tensile performance of the materials. By combining this observation with the comparison of tensile strength and fracture elongation for different composite materials depicted in Figure 5b, it becomes evident that, with the exception of the stearic acid-treated cotton stalk fiber composite (SA-CSF), all chemically treated cotton stalk fiber composites have exhibited an enhancement in their tensile performance. Compared with untreated composite materials, the tensile strength of CSFs treated with stearic acid decreases while the fracture elongation rate slightly increases. This is attributable to the formation of a stearic acid coating on the surface of CSFs through treatment. Although this coating reduces the mutual attraction between polymer molecular chains and CSFs, it also lubricates the interface between them, facilitating chain segment slip during stretching and promoting polymer molecular chain movement. Consequently, material tensile performance decreases, but the fracture elongation rate experiences a slight increase. The alkali-treated cotton stalk fiber (N-CSF) effectively eliminates impurities such as pectin and wax from the fiber surface, thereby exposing the primary mechanical support component, cellulose. This increase in cellulose content significantly enhances the tensile performance of the composite material. The silicon-treated cotton stalk fiber composite (S-CSF) also exhibits improved tensile properties, albeit to a lesser extent compared with N-CSF. This can be attributed to the formation of a uniform layer of polysiloxane coating on the surface of CSF by silane coupling agents, which enhances interface bonding strength between fibers and the PLA matrix. However, it does not effectively prevent separation at interfaces due to components such as lignin and pectin within CSFs that adversely affect material tensile performance. The performance of silicon-treated cotton stalk fiber composite (S-CSF) has also been improved, albeit not to the extent observed in N-CSF. This is because although the silane coupling agent forms a relatively uniform siloxane layer on the surface of the CSFs, enhancing the interfacial bonding strength between the fibers and the PLA matrix, effectively preventing interface separation, the presence of lignin, pectin, and other components within the CSFs also influences the tensile properties of the material. Surprisingly, the tensile performance of the alkali + silicon-treated cotton stalk fiber composite (NS-CSF) has been significantly enhanced. It combines the advantages of alkali treatment and silicon treatment. The alkali treatment removes a series of impurities, such as pectin, from the surface of the CSFs, increasing the cellulose content. However, this treatment may induce certain surface defects and flaws. The silane coupling agent treatment can effectively fill these microscopic defects on the surface of CSFs and form a uniform siloxane layer, thereby reducing interface defects and simultaneously enhancing interfacial bonding strength between fibers and matrix. This improved interfacial bonding strength effectively prevents interface separation or debonding, ultimately leading to an enhancement in tensile strength for the composite material. After comparing the flexural strength and impact strength of different materials, it was observed that the flexural strength of the SA-CSF component increased while the impact strength decreased. This can be attributed to a more uniform distribution and arrangement of stearic acid-coated CSFs within the PLA matrix compared with untreated fibers. The orderly fiber alignment facilitates silver line formation during bending experiments, reducing stress concentration and local plastic deformation in materials, thereby enhancing flexural strength. However, in notch impact testing, the regular fiber distribution makes it easier for materials to exhibit fracture paths under impacts, resulting in reduced impact strength. Similarly, the behavior of the S-CSF component is also observed in a similar manner. The presence of a silane film, generated by coating CSFs with a silane coupling agent, enhances the bonding strength between the fibers and the PLA matrix. However, while improving bending performance through enhanced fiber orientation regularity, it simultaneously results in a reduction in impact performance. The alkali treatment of N-CSF effectively removes impurities from its surface, resulting in an increased cellulose content and enhanced fiber surface roughness, as well as an increase in hydroxyl groups. This leads to a stronger interaction between the fiber surface and PLA matrix molecular chains, promoting mechanical anchoring points between fibers and the matrix while reducing stress concentration at the interface. Compared with S-CSF and N-CSF, it exhibits superior absorption and dispersion of external impact loads, thereby improving both the bending strength and impact strength of composite materials. The alkali and silane-treated cotton stalk fiber composite material (NS-CSF) combines the benefits of alkali treatment, while the subsequent silane treatment forms a protective layer on the rough fiber surface, thereby enhancing the interfacial bonding strength between fibers and PLA matrix, consequently improving its flexural performance. However, there exists a dual effect as the silane layer enhances fiber dispersion within the PLA matrix, resulting in a more regular orientation and orderly arrangement. Consequently, NS-CSF exhibits higher impact strength compared with UN-CSF but lower impact resistance than N-CSF.

### 3.5. Impact Fracture Surface Morphology Analysis of Composites

The fracture surfaces of the impact specimens were characterized and analyzed using a scanning electron microscope (SEM). The SEM images in Figure 6 depict the fracture surfaces of each composite. Figure 6a shows the microstructure of untreated cotton stalk fiber composites, while Figure 6b showcases those treated with stearic acid. Figure 6(c1,c2) exhibits those treated with alkali, whereas Figure 6(d1,d2) displays those treated with silane. Figure 6(e1–e3) demonstrates those subjected to alkali/silane treatment.

By comparison, it can be observed that the distribution of CSFs treated with stearic acid in the polymer matrix is uneven. Additionally, a layer of stearic acid coating on the surface of these fibers hinders their adhesion to the matrix, thus confirming our previous speculation [46,55]. Figure 5c1,c2 depict the impact fracture surfaces of alkali-treated cotton stalk fiber composites (N-CSF), revealing a rough surface with numerous mechanical anchor points on the CSFs. The effect of fiber plucking caused by impact is evident from the presence of rough impact surfaces and significant deformation surfaces [38,41,46,47,48]. The microstructure of impact fracture surfaces of cotton stalk fiber composite (S-CSF) treated with a silane coupling agent is depicted in Figure 5d1,d2. It can be observed that the application of a silane film effectively enhances the bonding ability between CSFs and the matrix. Linear polymer connections are also present between protruding fibers and the PLA matrix on the impact surface. Similarly, polymer adhesion is observed on the fracture surface of composites treated with an alkali + silane coupling agent, which increases the interaction force between the cotton stalk fiber surface and PLA matrix due to an increase in hydroxyl carboxyl groups on fiber surfaces after alkali treatment. The most noticeable manifestation is that there is an increased thickness and number of bonding lines between CSFs and the matrix, indicating a stronger adhesive effect [44,45,56].

### 3.6. Analysis of the Dynamic Mechanical Properties of Composites

The response of viscoelasticity and mechanical properties of materials to temperature changes was evaluated using dynamic mechanical analysis (DMA). Figure 7 illustrates the variations in storage modulus (Figure 7(a1,a2)), loss modulus (Figure 7(b1,b2)), tanδ (Figure 7c), and glass transition temperature (T_g_) (Figure 7d) of cotton stalk fiber composite materials after undergoing different chemical treatments under varying temperatures.

All composite materials exhibit a consistent trend in the variation in their storage modulus under variable temperature conditions, characterized by stability at low temperatures followed by a sharp decrease in storage modulus at 60 °C. The highest storage modulus observed in N-CSF indicates excellent material rigidity conferred by the alkaline-treated cotton stalk fiber composite, enabling it to resist deformation and damage better when subjected to impact. Additionally, the enhanced fatigue resistance of the material allows it to maintain its structure and performance for longer periods under cyclic impact loading. This performance is primarily attributable to the increase in cellulose content resulting from the alkaline treatment, which facilitates synergistic interactions between the matrix polymer and the alkaline-treated CSFs. The effective improvement in mechanical properties further validates this assertion [23,41,57]. Both stearic acid-treated (SA-CSF) and silane-treated (S-CSF) cotton stalk fiber composites show an increase in storage modulus, and their storage modulus curves exhibit similar trends. This is attributable to the fact that both treatments involve coating the fiber surface, resulting in uniform fiber distribution and regular orientation. Consequently, the materials possess enhanced energy absorption capabilities, slowing down the propagation speed of impact loads and reducing stress levels in the materials. Moreover, the van der Waals forces between the silane coating and the matrix are stronger than those between the stearic acid coating and the matrix, leading to an overall higher storage modulus in S-CSF compared with SA-CSF [45,47,58]. The storage modulus of the composite material treated with both alkali and silane (NS-CSF) is lower than that of other composite materials. This decrease in stiffness makes the material more prone to bending and deformation under impact loading. Furthermore, as depicted in Figure 7b2, NS-CSF exhibits a decrease in loss modulus at lower temperatures, indicating that its viscoelastic properties under dynamic loading are more susceptible to temperature variations. This susceptibility can be attributed to the broader and more uniform distribution of alkali/silane-treated CSFs in the composite material compared with solely alkali-treated fibers. Additionally, the surface silane coating enhances the entanglement between the CSFs and the polymer chains. This entanglement results in numerous polymer chain interlacings, forming discontinuous crystalline regions within the polymer matrix. Under certain frequency and strain conditions, such interfacial relaxation and increased intermolecular friction occur, rendering the material more susceptible to temperature-induced reductions in loss modulus. The peak value of tan δ represents the material’s loss factor, corresponding to its T_g_, which signifies the material’s deformation behavior during practical applications. As shown in Figure 7c, NS-CSF exhibits the highest peak, attributable to the discontinuity of the PLA crystalline regions, resulting in molecular chain segment movement at lower temperatures. Additionally, Figure 7d illustrates that NS-CSF has the lowest glass transition temperature, attributable to structural relaxation within the interface regions of the composite material, making the molecular chain segments more susceptible to temperature variations under relative frequency-induced stress.

### 3.7. Thermal Stability Analysis of the Composites

The Vicat softening points of composite materials composed of differently chemically treated CSFs were measured, and their thermal degradation behavior under a nitrogen atmosphere was analyzed using a thermogravimetric analyzer. Figure 8a presents a comparison of the Vicat softening points of the composite materials. The TG and DTG curves of the composite materials are shown in Figure 8(b1,b2,c1,c2), respectively, while Figure 8d depicts a comparative chart of the initial degradation temperature (T_d initial_), the temperature at 25% thermal degradation (T_d 25%_), and the temperature of the maximum degradation rate (T_d max_) for each composite material.

The Vicat softening point of untreated cotton stalk fiber composite (UN-CSF) is 72.6 °C, while that of stearic acid-treated cotton stalk fiber composite (SA-CSF) is 66.9 °C, indicating a decrease of 5.7 °C. This reduction is attributable to stearic acid, a low molecular weight fatty acid commonly used as a plasticizer, which interacts with polymer molecules, reducing their mutual attraction and making polymer chains more susceptible to flow at high temperatures. Additionally, stearic acid coating on cotton stalk fiber surfaces reduces electrostatic attraction and van der Waals forces between polymer chains and fiber surfaces, further facilitating polymer chain movement and lowering the Vicat softening point. The Vicat softening point of alkali-treated cotton stalk fiber composite (N-CSF) is 73.9 °C, which is 1.3 °C higher than UN-CSF. Alkali treatment removes impurities such as pectin, wax, and lignin from the fiber surface, increasing the number of available -OH groups. This strengthens the electrostatic attraction and van der Waals forces between polymer chains and CSFs, limiting molecular chain mobility at high temperatures and raising the composite material’s Vicat softening point. The Vicat softening point of silane-treated cotton stalk fiber composite (S-CSF) is 73 °C, a 0.4 °C increase compared with UN-CSF. Silane coupling agents are surfactants used to improve the interface compatibility between fibers and polymers. When CSFs are compounded with PLA, silane coupling agents promote bonding between them. Chemical modification of the cotton stalk fiber surface by silane treatment enhances interaction with PLA material. This improved interaction increases interfacial bonding, preventing interface separation or delamination, thereby enhancing the heat resistance of the composite. Additionally, silane treatment imparts hydrophobicity to the cotton stalk fiber surface, reducing heat conduction between fibers and PLA and minimizing intermolecular interactions, thus increasing the material’s Vicat softening point. The Vicat softening point of alkali/silane-treated cotton stalk fiber composite (NS-CSF) is 76.4 °C, a 3.8 °C increase compared with UN-CSF. The temperature increase is more significant compared with individual alkali or silane treatment, as alkali treatment removes impurities and increases the number of available -OH groups, while silane coupling agent application sites increase. The uniform silane coating on the fiber surface further reduces heat conduction between fibers and PLA, decreases intermolecular interactions, and strengthens the interaction between silane-coated fiber surfaces and PLA, resulting in improved interfacial bonding and higher Vicat softening point. Moreover, different chemical treatments of CSFs can affect the crystalline properties of composite materials, resulting in variations in the Vicat softening point. Based on the TG curves (Figure 8b1), it is evident that all composite materials undergo a two-step degradation process. Analyzing the initial decomposition temperature, temperature at 25% thermal degradation rate, and temperature of maximum degradation rate depicted in Figure 8d, it can be observed that the thermal stability of stearic acid-treated cotton stalk fiber composite (SA-CSF) exhibits a slightly lower level compared with untreated cotton stalk fiber composite (UN-CSF), with temperatures decreasing by approximately 1–2 °C. This decrease can be attributed to the relatively weaker thermal stability of the stearic acid film coating on the surface of CSFs. Stearic acid is susceptible to oxidation reactions at elevated temperatures, resulting in its decomposition into shorter-chain fatty acids, water, and other decomposition products. These decomposition products may react with the polymer matrix, thereby accelerating its thermal degradation [45,59,60]. The thermal stability of other chemically treated cotton stalk fiber composites has been enhanced. Specifically, alkali treatment of CSFs effectively removes easily decomposable pectin and hemicellulose components while retaining a higher proportion of stable cellulose, thereby reducing fiber decomposition and flammability. This reduction leads to a decrease in thermal decomposition reactions at elevated temperatures, consequently enhancing the overall thermal stability of the composite material. On the other hand, the presence of a silicon film on the surface of silicon-treated CSFs, due to its unique O-Si-C structure, imparts greater stability compared with the C-C structure alone. As a result, it significantly improves the material’s thermal stability [44,48,49,60]. The cotton stalk fiber composite treated with alkali/silane (SN-CSF) undergoes an initial alkali treatment, which effectively removes the easily decomposable unstable hemicellulose and pectin components, thereby enhancing the surface roughness. Subsequent silicic acid treatment further improves the fiber’s thermal stability by protecting its surface defects and morphology with a silicon film. Consequently, when these fibers are incorporated into the composite material, they create a layered barrier effect that restricts heat penetration. As a result, SN-CSF exhibits significantly higher thermal stability compared with cotton stalk fiber composites treated solely with either alkali (N-CSF) or silane (S-CSF). By combining the results from Figure 8d regarding the initial decomposition temperature (T_d initial_), temperature at 25% thermal degradation (T_d 25%_), and maximum thermal degradation rate temperature (T_d max_) with those obtained from DTG analysis in Figure 8c2, it is evident that dual treatment of alkali and silane leads to the highest thermal decomposition temperature for NS-CSF.

### 3.8. DSC Analysis of the Composites

The melting and crystallization properties of cotton stalk fiber-reinforced composite materials after different chemical treatments were evaluated using differential scanning calorimetry (DSC). Figure 9 presents the first heating curve (a), cooling curve (b), and second heating curve (c) of various composite materials. Figure 9d illustrates the comparison of crystallinity between two heating cycles. Detailed thermal data can be found in Appendix A.

Compared with untreated cotton stalk fiber composite material (UN-CSF), all composite materials exhibited a decrease in glass transition temperature (T_g_). Notably, the stearic acid-treated cotton stalk fiber (SA-CSF) demonstrated the smallest reduction in T_g_ value. This can be attributed to the formation of a stearic acid film on the fiber surface through treatment, which effectively diminishes intermolecular interactions between polymer chains and facilitates their flow at elevated temperatures. This phenomenon aligns with the previously mentioned decline in Vicat softening point [47]. Furthermore, the crystallization temperature (T_c)_ of SA-CSF also decreased. This is attributable to the effective dispersion and alignment of stearic acid-treated CSFs within the polymer matrix, resulting in a more homogeneous distribution and orientation. Consequently, this facilitates the initiation of crystallization in the semicrystalline polypropylene composite material. The abundant and uniformly dispersed stearic acid-treated CSFs promote a uniform distribution of polypropylene, acting as nucleation sites for PLA crystallization. As a result, PLA crystals initiate at these sites, leading to overall crystal growth and thereby enhancing material quality by promoting PLA crystallization and improving overall crystallinity. The observed increase in SA-CSF’s crystallinity further supports this inference. The alkaline-treated CSFs, due to the removal of surface impurities, result in a roughened fiber surface with increased surface hydroxyl and carboxyl functional groups. This multifaceted effect enhances the interface compatibility with the matrix components, thereby bolstering the interaction forces with the polymer chains. Consequently, the composite material exhibits an elevated T_g_ in comparison to SA-CSF. However, owing to intensified interaction forces, the dispersion of CSFs within the polymer matrix is compromised, leading to an uneven distribution of crystallization nucleation sites such as those found in polypropylene. As a result, this non-uniform distribution engenders a reduction in overall material crystallinity [58]. In the cooling curves of the composite materials, it is observed that there is no significant change in the thermal crystallization temperature between the two cycles. The CSFs treated with silane coupling agent exhibit enhanced interaction forces with the polymer chains due to their uniquely coated surface with silane. However, the dispersion of fibers is superior to that of N-CSF. Consequently, CSFs can interact more effectively with polypropylene, facilitating the rapid generation of polypropylene microcrystals. This phenomenon promotes an increase in the crystallinity of the composite material [23,61]. Although the CSFs treated with alkali/silane are coated with a silane layer on the surface, the previous alkali treatment results in increased surface roughness of the fibers. This roughness adversely affects the dispersion and alignment of CSFs in the polymer matrix, leading to lower crystallinity compared with S-CSF but higher than N-CSF. Consequently, the enhancement of crystalline performance in composite materials is primarily attributable to the interaction between CSFs and the polymer matrix interface, as well as the dispersion and alignment of fibers within the composite matrix. Better dispersion and alignment of fibers result in improved crystallinity, whereas increased surface roughness and the presence of surface hydrogen and chemical bonds lead to stronger interactions with the matrix, thereby hindering the increase in crystallinity of the material [23]. Materials with high crystallinity often exhibit higher hardness and stiffness, as the crystalline regions typically enhance molecular alignment and order. Moderate crystallinity can improve the mechanical properties of a material by impeding stress transmission as resistance points to dislocations or cracks. For cotton stalk fiber-reinforced composite materials, the improvement in mechanical properties is a result of the combined effect of the matrix material and the fiber reinforcement. Increasing the roughness of the fiber surface enhances the interfacial bonding between the fiber and the matrix, preventing interfacial separation and strengthening the mechanical properties of the material.

### 3.9. Composites Contact Angle Analysis

The wettability of all composite materials was analyzed using a water contact angle tester. The comparison of water contact angles for the composite materials is shown in Figure 10. The water contact angle of untreated cotton stalk fiber composite material (UN-CSF) was measured to be 90.7°. For all chemically treated cotton stalk fiber composite materials, the water contact angles decreased. Among them, the composite material treated with stearic acid showed the largest decrease in contact angle. This is primarily due to the uniform distribution of stearic acid-coated CSFs in the PLA matrix, resulting in a smoother and more uniform surface of the composite material. Consequently, this increased the contact area of water on the surface, leading to a decrease in the water contact angle. Although alkali treatment may increase the roughness of the composite material surface, it also increases the hydroxyl (-OH) content on the surface of CSFs. This formation of more hydrogen bonding contact points enhances the surface activity of the composite material, making it easier for water molecules to interact with the surface, thereby reducing the contact angle [61]. Silane-treated CSFs, despite their surface coating of silane, which facilitates uniform dispersion of the fibers in the polymer matrix, exhibit increased hydrophobicity due to the hydrophobic nature of the silane compound coating. This hydrophobic surface reduces the contact of water molecules with the surface, resulting in an increase in the water contact angle compared with SA-CSF [45]. Similarly, in alkali/silane-treated CSFs, the surface initially exposed numerous hydrophilic hydroxyl groups upon alkali treatment. However, the hydrophobic silane coating masks this advantage, leading to increased water contact.

### 3.10. Moisture Absorption Analysis

The moisture absorption of the composite material was evaluated by conducting water absorption tests. A comparison of the moisture absorption of the composite materials is shown in Figure 11. In comparison, it can be observed that the SA-CSF component demonstrates superior moisture absorption characteristics. This is attributable to the presence of stearic acid, which forms a coating on the surface of cotton stalk fibers. Stearic acid, being a long-chain fatty acid, contains hydrophilic groups such as carboxyl groups in its molecular structure. These hydrophilic groups facilitate interactions with water molecules and attract them into the polylactic acid structure. Moreover, treatment with stearic acid effectively eliminates impurities protruding from cotton stalk fiber surfaces, enhancing their smoothness and providing an efficient pathway for water molecules to permeate into the material’s interior, thereby further augmenting its moisture absorption capability. Alkali treatment removes components such as pectin, hemicellulose, and lignin from the surface of the cotton stalk fibers, resulting in increased surface roughness and irregular and porous features, which hinder the entry of water molecules into the material, thus weakening its moisture absorption capability. While silane treatment results in a cleaner surface of the cotton stalk fibers, the hydrophobic nature of the silane layer on the surface impedes the infiltration of water molecules, resulting in a decrease in the moisture absorption performance of the S-CSF component. The NS-CSF component has the worst moisture absorption performance among all composite materials. This is because the cotton stalk fibers in this component undergo alkali treatment, which removes pectin, hemicellulose, lignin, and other components from the fiber surface. As a result, the roughness of the fiber surface increases and disrupts the channels for water molecules to penetrate. Furthermore, silane treatment is applied to cover a hydrophobic silicon oxide coating on the surface of these fibers, further repelling water molecule infiltration and reducing their moisture absorption performance even more. The NS-CSF component exhibits the poorest moisture absorption performance among all composite materials due to the alkali treatment of cotton stalk fibers, which eliminates pectin, hemicellulose, lignin, and other components from the fiber surface. Consequently, the increased roughness of the fiber surface disrupts water molecule penetration channels. Additionally, a silane treatment is applied to create a hydrophobic silicon oxide coating on these fibers’ surfaces, further deterring water infiltration and reducing their moisture absorption performance. The conclusion can be drawn that the enhancement of cotton stalk fiber surface polarity and improvement in overall fiber integrity and smoothness can significantly augment the moisture absorption performance of composite materials.

## 4. Conclusions

In conclusion, this study investigated the impact of various chemical treatments on cotton stalk fiber-reinforced composite materials. Four distinct chemical treatments were applied to prepare differently treated CSF: stearic acid treatment, alkali treatment, silane coupling agent treatment, and alkali/silane coupling agent treatment. These treated fibers were subsequently utilized as fillers for composite material fabrication via a melt extrusion method. Our findings reveal that augmenting the surface roughness of CSFs enhances the overall mechanical properties of the composite materials, albeit at the expense of compatibility between the fibers and the composite matrix. Smooth fiber surfaces facilitate compatibility with PLA, while the Si-O-C silane coating increases fiber regularity and interfacial interaction strength, thereby enhancing thermal stability. Notably, composite materials incorporating alkali/silane-treated CSFs exhibit optimal mechanical properties and thermal stability. Specifically, compared with untreated counterparts, these composites demonstrated improved tensile strength (7.84% increase), flexural strength (8.04% increase), and impact strength (3.42% increase). Moreover, the Vicat softening temperature increased by 3.8 °C. Additionally, better fiber dispersion and alignment within the matrix contribute to enhanced overall crystallinity, albeit at the expense of impact resistance. Nevertheless, modifying the surface roughness of reinforcing fibers presents a promising approach to mitigate this limitation. These findings underscore the potential of tailored chemical treatments in optimizing the performance of natural fiber-reinforced composite materials for various applications.

## Figures and Tables

**Figure 1 polymers-16-01641-f001:**
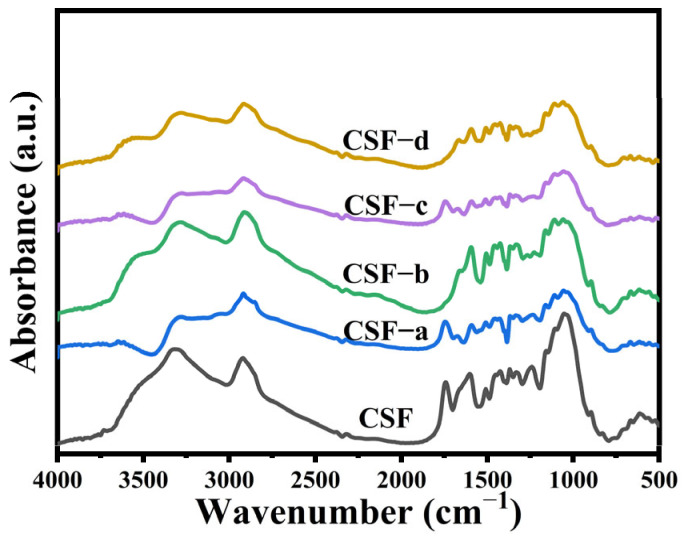
FTIR spectra of the cotton stalk fibers.

**Figure 2 polymers-16-01641-f002:**
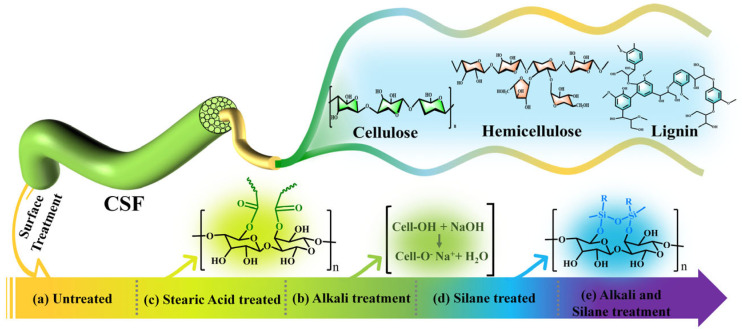
The reaction mechanism and morphological changes in cotton stalk fibers with different chemical treatments.

**Figure 3 polymers-16-01641-f003:**
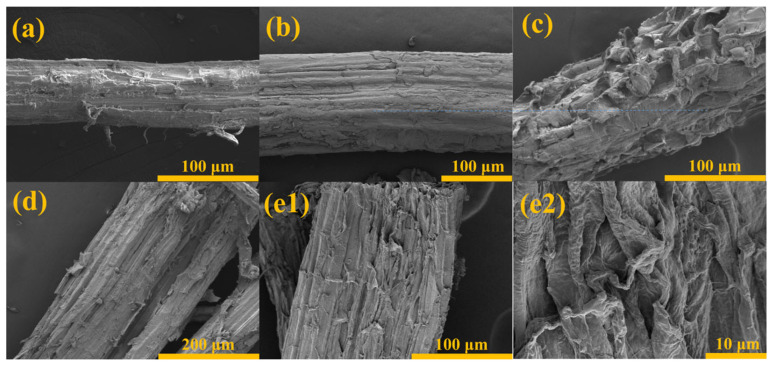
The SEM micrograph of (**a**)UN-CFS, (**b**) SA-CFS, (**c**) N-CFS, (**d**) S-CFS, (**e1**) NS-CFS, (**e2**) NS-CFS partial enlarged drawing.

**Figure 4 polymers-16-01641-f004:**
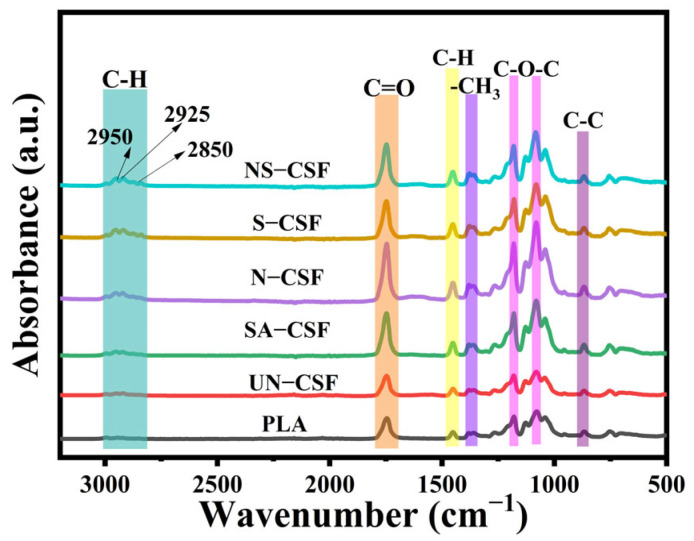
The FT-IR curves of the composites.

**Figure 5 polymers-16-01641-f005:**
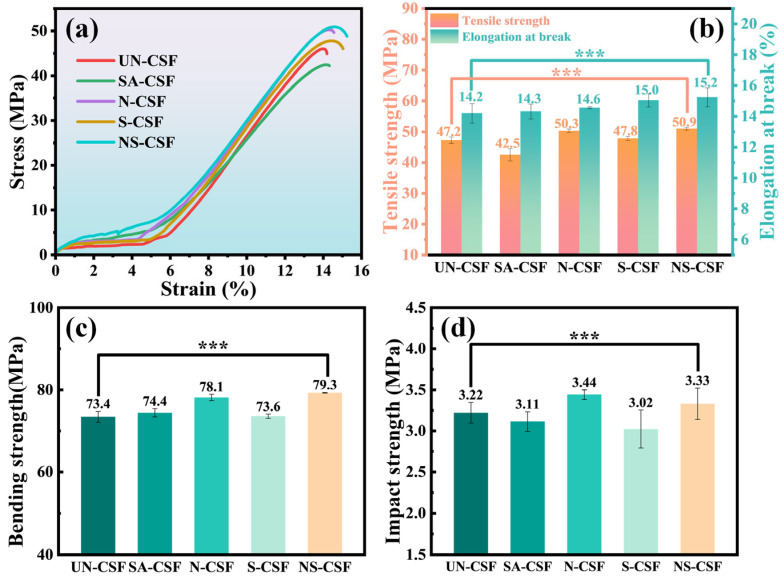
The mechanical properties of the composite material: (**a**) Tensile stress–strain curve, (**b**) Tensile strength and Elongation at break, (**c**) Bending strength, (**d**) Impact strength. *n* = 5, ***, *p* < 0.001.

**Figure 6 polymers-16-01641-f006:**
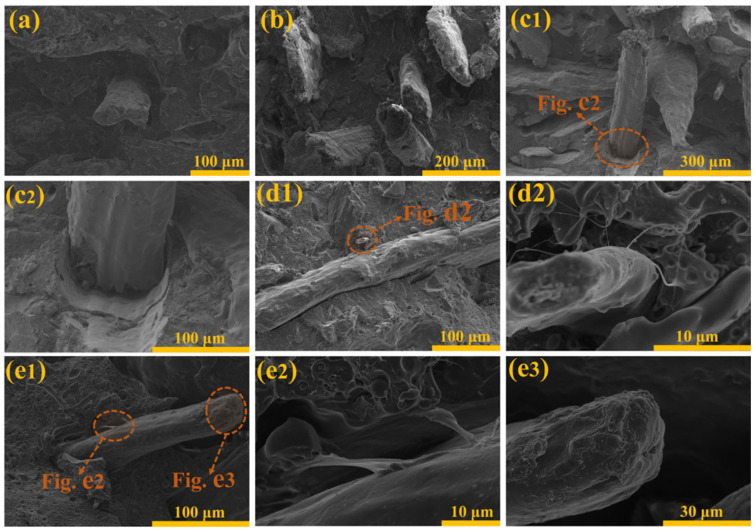
The SEM micrograph of (**a**) UN-CSF, (**b**) SA-CSF, (**c1**) N-CSF, (**c2**) N-CSF partial enlarged drawing, (**d1**) S-CSF, (**d2**) S-CSF partial enlarged drawing, (**e1**) NS-CSF partial enlarged drawing, (**e2**,**e3**) NS-CSF partial enlarged drawing.

**Figure 7 polymers-16-01641-f007:**
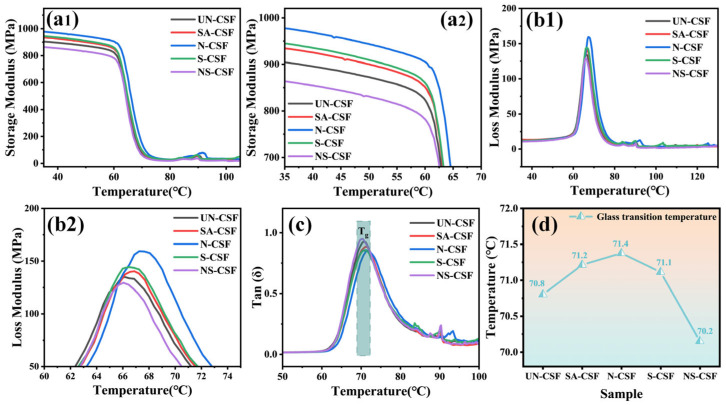
Dynamic mechanical curve: (**a1**) storage modulus, (**a2**) storage modulus amplification curve, (**b1**) loss modulus, (**b2**) loss modulus amplification curve, (**c**) tan δ, (**d**) Glass transition temperature.

**Figure 8 polymers-16-01641-f008:**
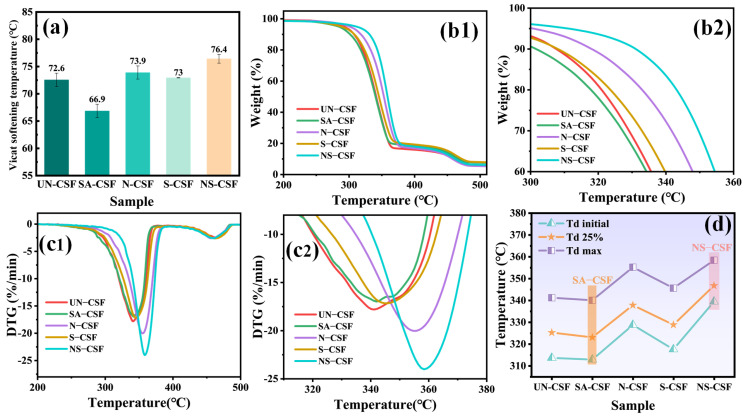
Comparison of Thermal Stability of Composites: (**a**) Vicat softening temperature, (**b1**) TG, (**b2**) TG amplification curve, (**c1**) DTG, (**c2**) DTG amplification curve, (**d**) T_d initial_ + T_d 25%_ + T_d max_.

**Figure 9 polymers-16-01641-f009:**
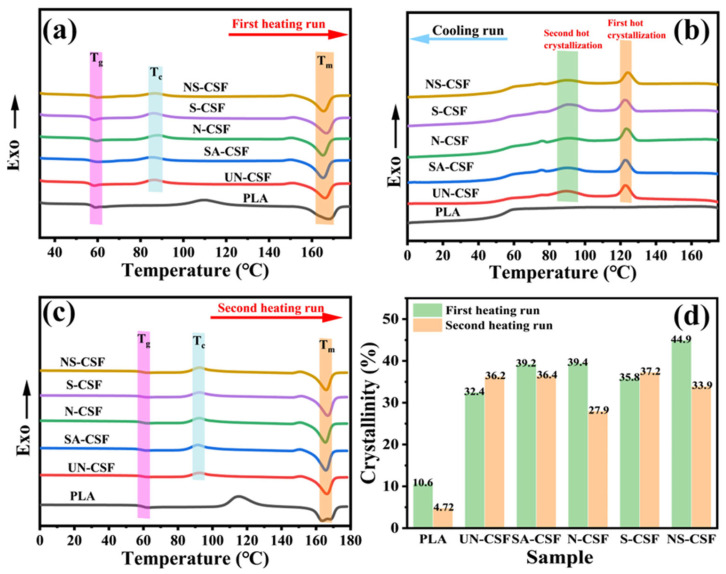
DSC Thermograms: (**a**) first heating run, (**b**) cooling run, (**c**) second heating run, and (**d**) crystallinity during two heating cycles.

**Figure 10 polymers-16-01641-f010:**
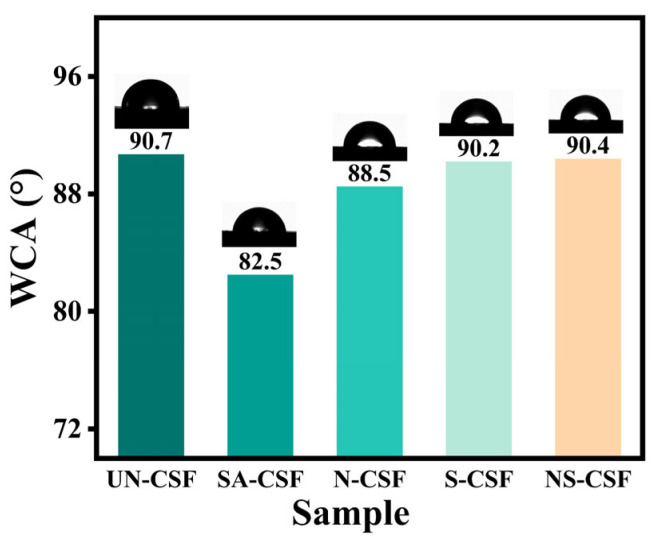
Water Contact Angle of Composites.

**Figure 11 polymers-16-01641-f011:**
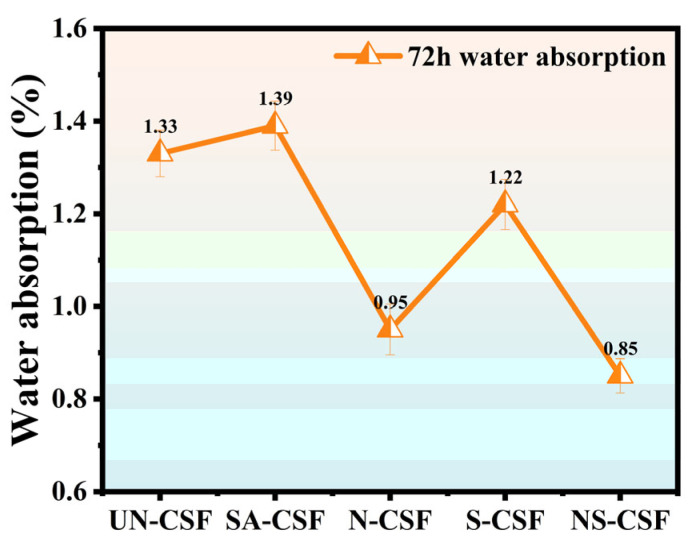
Moisture Absorption of Composites.

**Table 1 polymers-16-01641-t001:** The composition table for composite material.

Composites	PLA (wt%)	PP (wt%)	PP-g-MAH (wt%)	CSF (wt%)	Treatment Methods
UN-CSF	70	5	5	20	untreated
SA-CSF	70	5	5	20	stearic acid treatment
N-CSF	70	5	5	20	Alkaline treatment
S-CSF	70	5	5	20	Silane treatment
NS-CSF	70	5	5	20	Alkali/silane treatment

**Table 2 polymers-16-01641-t002:** Chemical composition of cotton stalk fibers with different chemical treatments.

Sample	Treatment Methods	Cellulose (w%)	Lignin (w%)	Hemicellulose (w%)
CSF	untreated	39.40	18.68	22.71
CSF-a	stearic acid treatment	43.75	20.24	17.83
CSF-b	Alkaline treatment	55.25	15.81	20.95
CSF-c	Silane treatment	42.97	22.15	21.30
CSF-d	Alkali/silane treatment	51.18	16.14	20.71

## Data Availability

Data are contained within the article.

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
