# Peer review of "Effect of Chemical Treatment of Cotton Stalk Fibers on the Mechanical and Thermal Properties of PLA/PP Blended Composites"

_polymers, 2024, doi:10.3390/polym16121641_

Round 1
Reviewer 1 Report
Comments and Suggestions for Authors
The first time an acronym appears in the text, it must be spelled out.
What particle size was used? Did it pass through a 60-mesh sieve?
You did not use fibers; you used particles.
After the stearic acid treatment, were the particles cleaned up?
Table 1 presents some results. It should be included in Section 3: Results and Discussion.
The methodology for all characterizations must be presented in detail.
The description of Figure 1 should be comprehensive to improve understanding, including CSF-a, b, c, and d.
The FT-IR analysis results must be compared with those presented in Table 1.
How many specimens were used in each treatment for the mechanical analyses?
It is necessary to present a statistical analysis to explain the mechanical tests.
How were the crystallinity calculations performed? The methodology must be presented in the Materials and Methods section.
All graphs must maintain the color standard for each treatment.
Author Response
Response to Reviewers
Dear Reviewers:
Thanks a lot for your constructive comments and suggestions about our manuscript. Those comments are valuable for improving the quality of our paper. We have substantially revised our manuscript to address your concerns and meeting the high standards of Polymers. The comments were carefully addressed below and the involved changes have been highlighted (yellow) in the revised manuscript and Supplementary Information.
To Reviewer #1:
- The first time an acronym appears in the text, it must be spelled out.
My answer according to this question: Thank you for your advice. According to your suggestion, we have spelled out the full form of the acronym upon its first occurrence.
- What particle size was used? Did it pass through a 60-mesh sieve?
My answer according to this question: Thank you very much for your valuable suggestions. In the current work, we sieved the cotton stem fibers using a mesh size of 40-60 mesh to obtain fibers within the 40-60 mesh range, and the cotton stem fibers unable to pass through the 60-mesh sieve.
- You did not use fibers; you used particles.
My answer according to this question: Thank you very much for your valuable suggestions. We dispersed cotton stalk fibers, which resemble carbon fibers, within the matrix of the composite material. Their fibrous nature can also be observed through the microstructure (Figure. 3 and Figure. 6), hence they are referred to as fibers.
- After the stearic acid treatment, were the particles cleaned up?
My answer according to this question: Many appreciations to the reviewers for the valuable suggestion. We washed the cotton stalk fibers treated with stearic acid using deionized water after the stearic acid treatment to remove remaining stearic acid. The specific experimental procedures will be described in the experimental section, and the changes made have been highlighted (yellow) in the revised manuscript.
Detailed revisions can be found on page 3, lines 111-114
Immerse cotton stalk fibers in a 1w% solution of stearic acid alcohol at room temperature for 2 hours, followed by filtration, remove residual stearic acid from the sur-face of the fibers using deionized water, followed by drying treatment.
- Table 1 presents some results. It should be included in Section 3: Results and Discussion.
My answer according to this question: Thank you very much for your valuable suggestions. According to your suggestion, We have incorporated the data results from Table 1 into the "3.1 Infrared Spectroscopic Analysis of Cotton Stalk Fibers after Different Chemical Treatments" section.
Detailed revisions can be found on
3.1 Infrared Spectroscopic Analysis of Cotton stalk fibers after Different Chemical Treatments
The CSF are primarily composed of three natural polymers: cellulose, hemicellulose, and lignin[1]. The surface structure of CSF following different chemical treatments was characterized using Fourier transform infrared spectroscopy (FTIR), as illustrated in Figure 1. Combining the results of the compositional analysis of cotton stalk fibers under different treatments, as shown in Table 1, a comprehensive discussion of the effects of various chemical treatments on cotton stalk fibers is presented.
Table.2 Chemical composition of cotton stalk fibers with different chemical treatments
|
Sample |
Treatment methods |
Cellulose (w%) |
Lignin (w%) |
Hemicellulose (w%) |
|
CSF |
untreated |
39.40 |
18.68 |
22.71 |
|
CSF-a |
stearic acid treatment |
43.75 |
20.24 |
17.83 |
|
CSF-b |
Alkaline treatment |
55.25 |
15.81 |
20.95 |
|
CSF-c |
Silane treatment |
42.97 |
22.15 |
21.30 |
|
CSF-d |
Alkali/silane treatment |
51.18 |
16.14 |
20.71 |
In the infrared spectra of CSF, a prominent peak at 3318.98 cm-1 corresponding to the stretching vibration of hydroxyl groups (-OH) was observed. Additionally, a peak indicative of the presence of carbon-hydrogen bonds (C-H) was observed at 2921.61 cm-1. The peak detected at 1054.22 cm-1 corresponds to the vibration of C-O-C in cellulose and hemicellulose, indicating the presence of polysaccharide chains in CSF.[2] Furthermore, the presence of benzene ring structures in CSF led to characteristic peaks associated with aromatic rings in the range of 1241 to 1603 cm-1.[3] The infrared peak corresponding to the hydroxyl groups (3318.98 cm-1) of CSF treated with stearic acid (CSF-a) showed a reduction. This reduction is likely due to a chemical reaction between stearic acid and the hydroxyl groups on the CSF, resulting in a decrease in the hydroxyl content on the fiber surface and the formation of ester bonds, thereby enhancing the infrared peak representing carbon-oxygen bonds (C=O) at 1748 cm-1. Additionally, the presence of stearic acid's eighteen-carbon alkane chain resulted in the encapsulation of the fiber surface, weakening the infrared peak associated with hydroxyl groups. The infrared spectrum of CSF treated with alkali (CSF-b) reveals an enhancement in the infrared peak attributed to the stretching vibration of carbon-hydrogen bonds (2921.61 cm-1) and the hydroxyl groups represented at 3318.98 cm-1. This enhancement is attributed to the alkali treatment's ability to remove pectin, polysaccharides, and other components from the surface of the CSF. Additionally, a decrease in the intensity of characteristic peaks related to aromatic groups, observed between 1241 and 1603 cm-1, is noted. This decrease is attributed to the removal of lignin components by the alkali treatment, resulting in surface roughening and the exposure of a large number of cellulose carbon-hydrogen bonds and hydroxyl groups.[4-8] The treatment of CSF with silane coupling agent (KH550) (CSF-c) leads to a reduction in the intensity of the hydroxyl group peak at 3318.98 cm-1, attributed to the effect of the silane chain. Additionally, an enhancement in the peaks corresponding to the Si-O-C functional group is observed at 1054 cm-1 and 1105 cm-1. In comparison to CSF treated solely with silane, those subjected to a combined treatment of alkali and silane (CSF-d) exhibit a strengthened hydroxyl peak at 3318.98 cm-1, albeit lower than that of alkali-treated fibers. Furthermore, a decrease in the peaks representing specific functional groups of lignin is noticed in the range of 1241 to 1603 cm-1, This phenomenon arises from the removal of lignin and pectin components due to alkali treatment, exposing a considerable number of hydroxyl groups. Subsequent silane treatment results in a reaction with these exposed hydroxyl groups, leading to a reduction in their peak intensity. [9-11]
Figure 1. FTIR spectra of the cotton stalk fibers.
- The methodology for all characterizations must be presented in detail.
My answer according to this question: Thank you very much for your valuable suggestions. According to your suggestion, we have included the specific experimental methods for characterization in the "2.4 Characterization studies" section. Detailed revisions can be found on page 4-5, lines 147-206:
2.4.1. Infrared spectroscopy analysis
The sample was subjected to chemical analysis using an Fourier-transform infrared spectrometer (FTIR) spectrometer (Thermo Scientific Nicolet IS20). The spectra were recorded by performing 32 scans over the wavenumber range of 400-4000 cm-1.
2.4.2. Heat resistance
The Vicat softening temperature (VST) of the bio-composite materials was evaluated using an HDT-Vicat tester in accordance with ISO 306:2004 standards.
2.4.3 Mechanical properties
The evaluation of the mechanical characteristics of both PLA and composites was carried out using a Universal Testing Machine. Tensile tests were performed following the ISO 527-2 standard. The impact strength was evaluated by employing notched specimens derived from the composites, and the assessments were conducted using an impact testing apparatus. Five parallel samples were selected for each test to ensure the representativeness of the data, Subsequently, the mean values obtained from these repetitions were documented along with their respective standard deviations.
2.4.4. Morphological studies
Investigate the microstructure of composite materials utilizing a Zeiss Sigma 300 scanning electron microscope (SEM) manufactured by Zeiss, Germany. Before conducting SEM analysis, a fine layer of gold was deposited onto the sample surfaces using a sputtering device to enhance conductivity and reduce charging phenomena. Thus,capturing high-resolution insights into the surface morphology of the samples.
2.4.5. Differential Scanning Calorimetry
The melting and crystallization characteristics of the specimens were investigated utilizing differential scanning calorimetry (DSC) with a TA DSC250, USA. Within the temperature range of -25 °C to 180 °C, the specimens were subjected to thermal cycling in a nitrogen gas environment. Calculate the crystallization rate utilizing the subsequent formula:
where and represent the respective enthalpies of melting and crystallization for the specimens. where and respectively denote the enthalpy change upon complete crystallization of PLA and PP, measured as 93.6 J/g and 188.9 J/g.
2.4.6. Thermal properties
To assess the thermal stability of the composites, executed thermogravimetric analysis (TGA) utilizing a German Netzsch TG 209 F3 Tarsus thermogravimetric analyzer. a 5 mg sample was placed into the instrument, and the heating temperature range was set from 30 °C to 600 °C.
2.4.7. Dynamic mechanical analysis
The dynamic mechanical properties of the composite materials were examined utilizing a PerkinElmer 8000 Dynamic Mechanical Analysis (DMA) instrument equipped with a single cantilever fixture.
2.4.8. Wettability analysis
The determination of the samples' hydrophilic and hydrophobic characteristics was carried out utilizing a Goniometer (Rigaku, Japan). Reposition the specimens onto a levelled platform and conduct surface analysis by means of water droplet examination. Reposition the specimens onto a levelled platform and conduct surface analysis by means of water droplet examination. The droplet deposition process was recorded. The contact angle data of the sample surface was calculated using software (laboratory desktop software Lab-Desk).
2.4.9. Moisture absorption analysis
According to the GB/T 1034-2008 standard for testing material moisture absorption, Five parallel samples were selected for each test to ensure the representativeness of the data, Subsequently, the mean values obtained from these repetitions were documented along with their respective standard deviations.
2.4.10. Statistical analysis
The basic parameters were visualized using Origin 2023b software (OriginLab Corporation, Northampton, MA, USA). Statistical analysis was conducted using the SPSS 25.0 software package (IBM Corp., Armonk, NY, USA). Data were presented as mean ± standard deviation (SD), and differences in experimental data were assessed by one-way analysis of variance (ANOVA) to determine statistical significance. A p-value less than 0.05 was considered statistically significant.
- The description of Figure 1 should be comprehensive to improve understanding, including CSF-a, b, c, and d.
My answer according to this question: Thank you very much for your valuable suggestions. According to your suggestion, We have expanded the description of Figure 1, providing a detailed discussion on the effects of chemical treatment on cotton stalk fibers by combining component analysis of the cotton stalk fibers with infrared spectroscopic analysis.Thanks again for your help. the involved changes have been highlighted (yellow) in the revised manuscript.
Detailed revisions can be found on
3.1 Infrared Spectroscopic Analysis of Cotton stalk fibers after Different Chemical Treatments
The CSF are primarily composed of three natural polymers: cellulose, hemicellulose, and lignin[1]. The surface structure of CSF following different chemical treatments was characterized using Fourier transform infrared spectroscopy (FTIR), as illustrated in Figure 1. Combining the results of the compositional analysis of cotton stalk fibers under different treatments, as shown in Table 1, a comprehensive discussion of the effects of various chemical treatments on cotton stalk fibers is presented.
Table.2 Chemical composition of cotton stalk fibers with different chemical treatments
|
Sample |
Treatment methods |
Cellulose (w%) |
Lignin (w%) |
Hemicellulose (w%) |
|
CSF |
untreated |
39.40 |
18.68 |
22.71 |
|
CSF-a |
stearic acid treatment |
43.75 |
20.24 |
17.83 |
|
CSF-b |
Alkaline treatment |
55.25 |
15.81 |
20.95 |
|
CSF-c |
Silane treatment |
42.97 |
22.15 |
21.30 |
|
CSF-d |
Alkali/silane treatment |
51.18 |
16.14 |
20.71 |
In the infrared spectra of CSF, a prominent peak at 3318.98 cm-1 corresponding to the stretching vibration of hydroxyl groups (-OH) was observed. Additionally, a peak indicative of the presence of carbon-hydrogen bonds (C-H) was observed at 2921.61 cm-1. The peak detected at 1054.22 cm-1 corresponds to the vibration of C-O-C in cellulose and hemicellulose, indicating the presence of polysaccharide chains in CSF.[2] Furthermore, the presence of benzene ring structures in CSF led to characteristic peaks associated with aromatic rings in the range of 1241 to 1603 cm-1.[3] The infrared peak corresponding to the hydroxyl groups (3318.98 cm-1) of CSF treated with stearic acid (CSF-a) showed a reduction. This reduction is likely due to a chemical reaction between stearic acid and the hydroxyl groups on the CSF, resulting in a decrease in the hydroxyl content on the fiber surface and the formation of ester bonds, thereby enhancing the infrared peak representing carbon-oxygen bonds (C=O) at 1748 cm-1. Additionally, the presence of stearic acid's eighteen-carbon alkane chain resulted in the encapsulation of the fiber surface, weakening the infrared peak associated with hydroxyl groups. The infrared spectrum of CSF treated with alkali (CSF-b) reveals an enhancement in the infrared peak attributed to the stretching vibration of carbon-hydrogen bonds (2921.61 cm-1) and the hydroxyl groups represented at 3318.98 cm-1. This enhancement is attributed to the alkali treatment's ability to remove pectin, polysaccharides, and other components from the surface of the CSF. Additionally, a decrease in the intensity of characteristic peaks related to aromatic groups, observed between 1241 and 1603 cm-1, is noted. This decrease is attributed to the removal of lignin components by the alkali treatment, resulting in surface roughening and the exposure of a large number of cellulose carbon-hydrogen bonds and hydroxyl groups.[4-8] The treatment of CSF with silane coupling agent (KH550) (CSF-c) leads to a reduction in the intensity of the hydroxyl group peak at 3318.98 cm-1, attributed to the effect of the silane chain. Additionally, an enhancement in the peaks corresponding to the Si-O-C functional group is observed at 1054 cm-1 and 1105 cm-1. In comparison to CSF treated solely with silane, those subjected to a combined treatment of alkali and silane (CSF-d) exhibit a strengthened hydroxyl peak at 3318.98 cm-1, albeit lower than that of alkali-treated fibers. Furthermore, a decrease in the peaks representing specific functional groups of lignin is noticed in the range of 1241 to 1603 cm-1, This phenomenon arises from the removal of lignin and pectin components due to alkali treatment, exposing a considerable number of hydroxyl groups. Subsequent silane treatment results in a reaction with these exposed hydroxyl groups, leading to a reduction in their peak intensity. [9-11]
Figure 1. FTIR spectra of the cotton stalk fibers.
- The FT-IR analysis results must be compared with those presented in Table1.
My answer according to this question: Thank you very much for your valuable suggestions. According to your suggestion, We have incorporated the data results from Table 1 into the “3.1 Infrared Spectroscopic Analysis of Cotton Stalk Fibers after Different Chemical Treatments” section, providing a detailed discussion on the effects of chemical treatment on cotton stalk fibers by combining component analysis of the cotton stalk fibers with infrared spectroscopic analysis. Thanks again for your help. the involved changes have been highlighted (yellow) in the revised manuscript.
Detailed revisions can be found on page 5-6, lines 208-251:
3.1 Infrared Spectroscopic Analysis of Cotton stalk fibers after Different Chemical Treatments
The CSF are primarily composed of three natural polymers: cellulose, hemicellulose, and lignin[1]. The surface structure of CSF following different chemical treatments was characterized using Fourier transform infrared spectroscopy (FTIR), as illustrated in Figure 1. Combining the results of the compositional analysis of cotton stalk fibers under different treatments, as shown in Table 1, a comprehensive discussion of the effects of various chemical treatments on cotton stalk fibers is presented.
Table.2 Chemical composition of cotton stalk fibers with different chemical treatments
|
Sample |
Treatment methods |
Cellulose (w%) |
Lignin (w%) |
Hemicellulose (w%) |
|
CSF |
untreated |
39.40 |
18.68 |
22.71 |
|
CSF-a |
stearic acid treatment |
43.75 |
20.24 |
17.83 |
|
CSF-b |
Alkaline treatment |
55.25 |
15.81 |
20.95 |
|
CSF-c |
Silane treatment |
42.97 |
22.15 |
21.30 |
|
CSF-d |
Alkali/silane treatment |
51.18 |
16.14 |
20.71 |
In the infrared spectra of CSF, a prominent peak at 3318.98 cm-1 corresponding to the stretching vibration of hydroxyl groups (-OH) was observed. Additionally, a peak indicative of the presence of carbon-hydrogen bonds (C-H) was observed at 2921.61 cm-1. The peak detected at 1054.22 cm-1 corresponds to the vibration of C-O-C in cellulose and hemicellulose, indicating the presence of polysaccharide chains in CSF.[2] Furthermore, the presence of benzene ring structures in CSF led to characteristic peaks associated with aromatic rings in the range of 1241 to 1603 cm-1.[3] The infrared peak corresponding to the hydroxyl groups (3318.98 cm-1) of CSF treated with stearic acid (CSF-a) showed a reduction. This reduction is likely due to a chemical reaction between stearic acid and the hydroxyl groups on the CSF, resulting in a decrease in the hydroxyl content on the fiber surface and the formation of ester bonds, thereby enhancing the infrared peak representing carbon-oxygen bonds (C=O) at 1748 cm-1. Additionally, the presence of stearic acid's eighteen-carbon alkane chain resulted in the encapsulation of the fiber surface, weakening the infrared peak associated with hydroxyl groups. The infrared spectrum of CSF treated with alkali (CSF-b) reveals an enhancement in the infrared peak attributed to the stretching vibration of carbon-hydrogen bonds (2921.61 cm-1) and the hydroxyl groups represented at 3318.98 cm-1. This enhancement is attributed to the alkali treatment's ability to remove pectin, polysaccharides, and other components from the surface of the CSF. Additionally, a decrease in the intensity of characteristic peaks related to aromatic groups, observed between 1241 and 1603 cm-1, is noted. This decrease is attributed to the removal of lignin components by the alkali treatment, resulting in surface roughening and the exposure of a large number of cellulose carbon-hydrogen bonds and hydroxyl groups.[4-8] The treatment of CSF with silane coupling agent (KH550) (CSF-c) leads to a reduction in the intensity of the hydroxyl group peak at 3318.98 cm-1, attributed to the effect of the silane chain. Additionally, an enhancement in the peaks corresponding to the Si-O-C functional group is observed at 1054 cm-1 and 1105 cm-1. In comparison to CSF treated solely with silane, those subjected to a combined treatment of alkali and silane (CSF-d) exhibit a strengthened hydroxyl peak at 3318.98 cm-1, albeit lower than that of alkali-treated fibers. Furthermore, a decrease in the peaks representing specific functional groups of lignin is noticed in the range of 1241 to 1603 cm-1, This phenomenon arises from the removal of lignin and pectin components due to alkali treatment, exposing a considerable number of hydroxyl groups. Subsequent silane treatment results in a reaction with these exposed hydroxyl groups, leading to a reduction in their peak intensity. [9-11]
Figure 1. FTIR spectra of the cotton stalk fibers.
- How many specimens were used in each treatment for the mechanical analyses?
My answer according to this question: Thank you very much for your valuable suggestions. We conducted mechanical property analysis using five or more test specimens. The detailed testing method is described in the "2.4.3 Mechanical properties" section of the article. Detailed revisions can be found on
Five parallel samples were selected for each test to ensure the representativeness of the data, Subsequently, the mean values obtained from these rep-etitions were documented along with their respective standard deviations.
- It is necessary to present a statistical analysis to explain the mechanical tests.
My answer according to this question: Many appreciations to the reviewers for the valuable suggestion. According to your suggestion, we added a statistical analysis section to the "2. Materials and Experimental" section, providing detailed descriptions of the statistical significance in Figure 5. The pertinent information is detailed within the manuscript, and the changes made have been highlighted (yellow) in the revised manuscript;
Detailed revisions can be found on page 5, lines 200-206
2.4.10. Statistical analysis
The basic parameters were visualized using Origin 2023b software (OriginLab Corporation, Northampton, MA, USA). Statistical analysis was conducted using the SPSS 25.0 software package (IBM Corp., Armonk, NY, USA). Data were presented as mean ± standard deviation (SD), and differences in experimental data were assessed by one-way analysis of variance (ANOVA) to determine statistical significance. A p-value less than 0.05 was considered statistically significant.
Figure.5 The mechanical properties of the composite material: (a) Tensile stress-strain curve (b) Tensile strength and Elongation at break (c) Bending strength (d) Impact strength. n = 5, ***, p < 0.001.
- How were the crystallinity calculations performed? The methodology must be presented in the Materials and Methods section.
My answer according to this question: Many appreciations to the reviewers for the valuable suggestion. According to your suggestion, we have added the method for calculating crystallinity to the "2. Materials and Experimental" section and the changes made have been highlighted (yellow) in the revised manuscript.
Detailed revisions can be found on page 4, lines 173-178:
Calculate the crystallization rate utilizing the subsequent formula:
where and represent the respective enthalpies of melting and crystallization for the specimens. where and respectively denote the enthalpy change upon complete crystallization of PLA and PP, measured as 93.6 J/g and 188.9 J/g.
- All graphs must maintain the color standard for each treatment.
My answer according to this question: Many appreciations to the reviewers for the valuable suggestion. According to your suggestion, we standardized the colors of the images in the article.

Reviewer 2 Report
Comments and Suggestions for Authors
The authors can find comments below:
1. The information provided supplementary information should be included in the main manuscript in the materials and methods sections.
2. The paper must be checked for the correctness of sentences. for example, on page 4 section 2.4, "In this study, we employed infrared analysis to investigate the surface structure of materials and evaluated their thermal performance through..."
"infrared analysis" seems confusing as the author performed infrared spectroscopy
3. Please check, if the provided data is FTIR transmittance or absorbance data in figure 1 and figure 4. The FTIR data characterization of cellulose materials (fibers) is usually observed by seeing the dips (peaks) for the OH, C-O, and CH stretching. However, if a particular chemical modification is performed then those chemical modifications can also be compared. For example, if there is NH2 modification over surface cellulose, someone can see the changes in transmittance dips (peaks) at 3300-3500 cm-1; C=O modification at 1690-1770 cm-1, and so on.
4. Were samples washed after performing the treatments?
Comments on the Quality of English Language
There are some typographical errors where the paper was not properly organized, especially in the material and method section.
Author Response
To Reviewer #2:
- The information provided supplementary information should be included in the main manuscript in the materials and methods sections.
My answer according to this question: Thank you very much for your valuable suggestions. According to your suggestion, we have added supplementary information to the Materials and Methods section of the paper.
- The paper must be checked for the correctness of sentences. for example, on page 4 section 2.4, "In this study, we employed infrared analysis to investigate the surface structure of materials and evaluated their thermal performance through..." "infrared analysis" seems confusing as the author performed infrared spectroscopy.
My answer according to this question: Thank you very much for your valuable suggestions. According to your suggestion, we have rectified the sections of the paper where errors were identified in the statements. At the same time, detailed descriptions of the methods for material performance testing and structure characterization were provided.
Detailed revisions can be found on page 4-5, lines 148-206
2.4.1. Infrared spectroscopy analysis
The sample was subjected to chemical analysis using an Fourier-transform infrared spectrometer (FTIR) spectrometer (Thermo Scientific Nicolet IS20). The spectra were recorded by performing 32 scans over the wavenumber range of 400-4000 cm-1.
2.4.2. Heat resistance
The Vicat softening temperature (VST) of the bio-composite materials was evaluated using an HDT-Vicat tester in accordance with ISO 306:2004 standards.
2.4.3 Mechanical properties
The evaluation of the mechanical characteristics of both PLA and composites was carried out using a Universal Testing Machine. Tensile tests were performed following the ISO 527-2 standard. The impact strength was evaluated by employing notched specimens derived from the composites, and the assessments were conducted using an impact testing apparatus. Five parallel samples were selected for each test to ensure the representativeness of the data, Subsequently, the mean values obtained from these repetitions were documented along with their respective standard deviations.
2.4.4. Morphological studies
Investigate the microstructure of composite materials utilizing a Zeiss Sigma 300 scanning electron microscope (SEM) manufactured by Zeiss, Germany. Before conducting SEM analysis, a fine layer of gold was deposited onto the sample surfaces using a sputtering device to enhance conductivity and reduce charging phenomena. Thus,capturing high-resolution insights into the surface morphology of the samples.
2.4.5. Differential Scanning Calorimetry
The melting and crystallization characteristics of the specimens were investigated utilizing differential scanning calorimetry (DSC) with a TA DSC250, USA. Within the temperature range of -25 °C to 180 °C, the specimens were subjected to thermal cycling in a nitrogen gas environment. Calculate the crystallization rate utilizing the subsequent formula:
where and represent the respective enthalpies of melting and crystallization for the specimens. where and respectively denote the enthalpy change upon complete crystallization of PLA and PP, measured as 93.6 J/g and 188.9 J/g.
2.4.6. Thermal properties
To assess the thermal stability of the composites, executed thermogravimetric analysis (TGA) utilizing a German Netzsch TG 209 F3 Tarsus thermogravimetric analyzer. a 5 mg sample was placed into the instrument, and the heating temperature range was set from 30 °C to 600 °C.
2.4.7. Dynamic mechanical analysis
The dynamic mechanical properties of the composite materials were examined utilizing a PerkinElmer 8000 Dynamic Mechanical Analysis (DMA) instrument equipped with a single cantilever fixture.
2.4.8. Wettability analysis
The determination of the samples' hydrophilic and hydrophobic characteristics was carried out utilizing a Goniometer (Rigaku, Japan). Reposition the specimens onto a levelled platform and conduct surface analysis by means of water droplet examination. Reposition the specimens onto a levelled platform and conduct surface analysis by means of water droplet examination. The droplet deposition process was recorded. The contact angle data of the sample surface was calculated using software (laboratory desktop software Lab-Desk).
2.4.9. Moisture absorption analysis
According to the GB/T 1034-2008 standard for testing material moisture absorption, Five parallel samples were selected for each test to ensure the representativeness of the data, Subsequently, the mean values obtained from these repetitions were documented along with their respective standard deviations.
2.4.10. Statistical analysis
The basic parameters were visualized using Origin 2023b software (OriginLab Corporation, Northampton, MA, USA). Statistical analysis was conducted using the SPSS 25.0 software package (IBM Corp., Armonk, NY, USA). Data were presented as mean ± standard deviation (SD), and differences in experimental data were assessed by one-way analysis of variance (ANOVA) to determine statistical significance. A p-value less than 0.05 was considered statistically significant.
3.
My answer according to this question: We appreciate the reviewers for the valuable suggestions. Thank you for your advice. According to your suggestion, we have modified the infrared spectra in Figure 1 and Figure 4 to display the data as absorbance data. Additionally, we have expanded the description of the infrared analysis.
Detailed revisions can be found on
3.1 Infrared Spectroscopic Analysis of Cotton stalk fibers after Different Chemical Treatments
The CSF are primarily composed of three natural polymers: cellulose, hemicellulose, and lignin[1]. The surface structure of CSF following different chemical treatments was characterized using Fourier transform infrared spectroscopy (FTIR), as illustrated in Figure 1. Combining the results of the compositional analysis of cotton stalk fibers under different treatments, as shown in Table 1, a comprehensive discussion of the effects of various chemical treatments on cotton stalk fibers is presented.
Table.2 Chemical composition of cotton stalk fibers with different chemical treatments
|
Sample |
Treatment methods |
Cellulose (w%) |
Lignin (w%) |
Hemicellulose (w%) |
|
CSF |
untreated |
39.40 |
18.68 |
22.71 |
|
CSF-a |
stearic acid treatment |
43.75 |
20.24 |
17.83 |
|
CSF-b |
Alkaline treatment |
55.25 |
15.81 |
20.95 |
|
CSF-c |
Silane treatment |
42.97 |
22.15 |
21.30 |
|
CSF-d |
Alkali/silane treatment |
51.18 |
16.14 |
20.71 |
In the infrared spectra of CSF, a prominent peak at 3318.98 cm-1 corresponding to the stretching vibration of hydroxyl groups (-OH) was observed. Additionally, a peak indicative of the presence of carbon-hydrogen bonds (C-H) was observed at 2921.61 cm-1. The peak detected at 1054.22 cm-1 corresponds to the vibration of C-O-C in cellulose and hemicellulose, indicating the presence of polysaccharide chains in CSF.[2] Furthermore, the presence of benzene ring structures in CSF led to characteristic peaks associated with aromatic rings in the range of 1241 to 1603 cm-1.[3] The infrared peak corresponding to the hydroxyl groups (3318.98 cm-1) of CSF treated with stearic acid (CSF-a) showed a reduction. This reduction is likely due to a chemical reaction between stearic acid and the hydroxyl groups on the CSF, resulting in a decrease in the hydroxyl content on the fiber surface and the formation of ester bonds, thereby enhancing the infrared peak representing carbon-oxygen bonds (C=O) at 1748 cm-1. Additionally, the presence of stearic acid's eighteen-carbon alkane chain resulted in the encapsulation of the fiber surface, weakening the infrared peak associated with hydroxyl groups. The infrared spectrum of CSF treated with alkali (CSF-b) reveals an enhancement in the infrared peak attributed to the stretching vibration of carbon-hydrogen bonds (2921.61 cm-1) and the hydroxyl groups represented at 3318.98 cm-1. This enhancement is attributed to the alkali treatment's ability to remove pectin, polysaccharides, and other components from the surface of the CSF. Additionally, a decrease in the intensity of characteristic peaks related to aromatic groups, observed between 1241 and 1603 cm-1, is noted. This decrease is attributed to the removal of lignin components by the alkali treatment, resulting in surface roughening and the exposure of a large number of cellulose carbon-hydrogen bonds and hydroxyl groups.[4-8] The treatment of CSF with silane coupling agent (KH550) (CSF-c) leads to a reduction in the intensity of the hydroxyl group peak at 3318.98 cm-1, attributed to the effect of the silane chain. Additionally, an enhancement in the peaks corresponding to the Si-O-C functional group is observed at 1054 cm-1 and 1105 cm-1. In comparison to CSF treated solely with silane, those subjected to a combined treatment of alkali and silane (CSF-d) exhibit a strengthened hydroxyl peak at 3318.98 cm-1, albeit lower than that of alkali-treated fibers. Furthermore, a decrease in the peaks representing specific functional groups of lignin is noticed in the range of 1241 to 1603 cm-1, This phenomenon arises from the removal of lignin and pectin components due to alkali treatment, exposing a considerable number of hydroxyl groups. Subsequent silane treatment results in a reaction with these exposed hydroxyl groups, leading to a reduction in their peak intensity. [9-11]
Figure 1. FTIR spectra of the cotton stalk fibers.
Detailed revisions can be found on
The intensity of the stretching vibration peak at 1748 cm-1 for C=O is correlated with the chemical treatment methods applied to CSF, ranked in order from strongest to weakest as follows: N-CSF > NS-CSF > S-CSF > SA-CSF > UN-CSF > PLA.[6] This correlation can be attributed to the interaction between CSF and PLA, which primarily occurs through bonding between hydroxyl groups (-OH) in CSF and carbonyl groups (C=O) and carboxyl groups (-COOH) in PLA. Due to the presence of pectin and wax materials on the surface of untreated CSF, a limited number of -OH groups are exposed, which can interact with the carbonyl (C=O) and carboxyl (-COOH) groups of PLA through hydrogen bonding and covalent bonding. In contrast, alkali-treated fibers (N-CSF) remove pectin and wax materials from the surface, increasing the available number of OH groups. Therefore, an increased exposure of -OH groups in the fibers enhances their interaction points with PLA's carbonyl (C=O) and carboxyl (-COOH) through hydrogen bonding and covalent bonding. The silanized cotton stalk fiber surface possesses various functional groups such as -OH and =NH, which can form hydrogen bonds and covalent bonds respectively with PLA's carbonyl (C=O) and carboxyl (-COOH). In SA-CSF, a layer of stearic acid coating on the surface results in stretching vibration peaks caused by its own C=O functional group being higher than that of untreated UN-CSF.
Figure.4 The FT-IR curves of the composites
- Were samples washed after performing the treatments?
My answer according to this question: Thank you very much for your valuable suggestions. Samples have been cleaned after treatment. The specific experimental procedures will be described in the experimental section, and the changes made have been highlighted (yellow) in the revised manuscript.
Detailed revisions can be found on page 3, lines 110-129:
2.2.1. Stearic Acid treatment
Immerse cotton stalk fibers in a 1w% solution of stearic acid alcohol at room tem-perature for 2 hours, followed by filtration, remove residual stearic acid from the sur-face of the fibers using deionized water, followed by drying treatment. These fibers shall be referred to as CSF-a.
2.2.2 Alkaline treatment
Soak cotton stalk fibers in a 0.5 w% NaOH aqueous solution and continuously agitate at room temperature for 24 hours. Subsequently, wash the resulting product with deionized water to achieve neutrality, and finally dry the product, designating it as CSF-b.
2.2.3 Silane treatment
The cotton stalk fibers were immersed in a 1 w% solution of γ-aminopropyltriethoxysilane (silane coupling agent KH-550), The solution was prepared by blending isopropanol and water in a ratio of 4:6, followed by agitation for 2 hours and subsequent filtration. Subsequently, the treated cotton stalk fibers were dried and designated as CSF-c.
2.2.4 Alkali/silane treatment.
The alkali-treated cotton stalk fibers were immersed in a 1w% solution of γ-aminopropyltriethoxysilane (silane coupling agent KH-550), which was prepared by mixing isopropanol and water in a ratio of 4:6. After stirring for 30 minutes, the fibers were further soaked for an additional 2 hours. Subsequently, the solution was filtered and the treated cotton stalk fibers were dried, resulting in their designation as CSF-d.
- There are some typographical errors where the paper was not properly organized, especially in the material and method section.
My answer according to this question: Thank you very much for your valuable suggestions. According to your suggestion, we have corrected the erroneous part in the narrative,especially in the material and method section.

Reviewer 3 Report
Comments and Suggestions for Authors
In this work different chemical treatment methods were employed to modify the surface of cotton stalk fibers, which were then utilized as fillers in composite materials. These treated fibers were incorporated into polylactic acid polypropylene melt blends using melt blending technique. The authors made an extensive experimental work and the whole work is interesting.
POINTS FOR IMPROVEMENT:
1. Please report the melt index & product of polypropylene
2. please report some technical data of PLA
3. Report some technical data on the fibers such as mechanical properties, etc
4. In table 1 report the pure fiber (without treatment) data. Also avoid using many decimal point. In my opinion one decimal point is enough.
5. A literature review made by the reviewer by using GOOGLESCHOLAR and authors keywords revealed about 35.000 references. Please, kindly respecify the keywords.
6. Please, add in the introduction a section about chemical treatment in fibers.
7. Please, elaborate more on the effect of crystallinity on the mechanical properties.
8. The literature review is incomplete. A detailed review on fibers chemical treatment could be found in Thapliyal, D.; Verma, S.; Sen, P.; Kumar, R.; Thakur, A.; Tiwari, A.K.; Singh, D.; Verros, G.D.; Arya, R.K. Natural Fibers Composites: Origin, Importance, Consumption Pattern, and Challenges. J. Compos. Sci. 2023, 7, 506. https://doi.org/10.3390/jcs7120506
9. Please, ellaborate more on moisture absorption which is a main effect of fibers chemical treatment.
Author Response
To Reviewer #3:
In this work different chemical treatment methods were employed to modify the surface of cotton stalk fibers, which were then utilized as fillers in composite materials. These treated fibers were incorporated into polylactic acid polypropylene melt blends using melt blending technique. The authors made an extensive experimental work and the whole work is interesting.
1.Please report the melt index & product of polypropylene
My answer according to this question: Thank you very much for your valuable suggestions. According to your suggestion, we have added the brand and melt index of the polypropylene in the "2.1. Materials" section.
Detailed revisions can be found on page 2-3, lines 102-104: PP (grade T30S) [MFR = 1.14 g/10 min (190 °C/2.16 Kg)], provided by Dushanzi Petro-chemical Company were utilized for this study.
- please report some technical data of PLA
My answer according to this question: Thank you very much for your valuable suggestions. According to your suggestion, we have added the technical data for polylactic acid in the "2.1. Materials" section.
Detailed revisions can be found on page 2, lines 101-102: The PLA (grade 4032D) [MFR = 7 g/10 min (210 °C/2.16 kg)], supplied by Nature Works LLC (Minnetonka, Minnesota, USA)
- Report some technical data on the fibers such as mechanical properties, etc.
My answer according to this question: Thank you very much for your valuable suggestions. According to your suggestion, we added the technical data for cotton stalk fibers in the "2.1. Materials" section.
Detailed revisions can be found on page 3, lines 104-105:
Cotton stalks was procured from the local market [Density 1.21g/cm3, Diameter 12–35 μm, Length 15–56 mm, Tensile Strength 287–597 MPa, Elongation at Break 2–10 %].
- In table 1 report the pure fiber (without treatment) data. Also avoid using many decimal point. In my opinion one decimal point is enough.
My answer according to this question: Thank you very much for your valuable suggestions. According to your suggestion, we have standardized the significant figures in Table 1.
Detailed revisions can be found on page 6, lines 248-249:
Table.2 Chemical composition of cotton stalk fibers with different chemical treatments
|
Sample |
Treatment methods |
Cellulose (w%) |
Lignin (w%) |
Hemicellulose (w%) |
|
CSF |
untreated |
39.40 |
18.68 |
22.71 |
|
CSF-a |
stearic acid treatment |
43.75 |
20.24 |
17.83 |
|
CSF-b |
Alkaline treatment |
55.25 |
15.81 |
20.95 |
|
CSF-c |
Silane treatment |
42.97 |
22.15 |
21.30 |
|
CSF-d |
Alkali/silane treatment |
51.18 |
16.14 |
20.71 |
- A literature review made by the reviewer by using GOOGLESCHOLAR and authors keywords revealed about 35.000 references. Please, kindly respecify the keywords.
My answer according to this question: Thank you very much for your valuable suggestions. According to your suggestion, we have modified the keywords of the article. The revised keywords are: polylactic acid; cotton stalk fibers; natural fiber reinforced composites; chemical treatment.
- Please, add in the introduction a section about chemical treatment in fibers.
My answer according to this question: Thank you very much for your valuable suggestions. According to your suggestion, we have added a description of fiber chemical treatment in the introduction section.
Detailed revisions can be found on page 2, lines 52-58:
Chemical processing involves immersing fiber straw, such as in water-soluble chemical solutions like potassium hydroxide, sodium hydroxide, and sulfuric acid, to selectively eliminate non-cellulosic components while retaining the functional cellulose constituents. Coupling agent treatment, waterproofing chemical modification, and heat treatment are employed to modify the morphology, roughness, and surface polarity of the fibers. [12, 13] This enhances compatibility and interactions between CSF and polymer matrices, ultimately leading to improved performance of resulting composite materials.
- Please, elaborate more on the effect of crystallinity on the mechanical properties.
My answer according to this question: Thank you very much for your valuable suggestions. According to your suggestion, we have added a discussion related to the mechanical properties of the material in the section explaining the changes in crystallinity.
Detailed revisions can be found
Materials with high crystallinity often exhibit higher hardness and stiffness, as the crystalline regions typically enhance molecular alignment and order. Moderate crystallinity can improve the mechanical properties of a material by impeding stress transmission as resistance points to dislocations or cracks. For cotton stalk fiber-reinforced composite materials, the improvement in mechanical properties is a result of the combined effect of the matrix material and the fiber reinforcement. Increasing the roughness of the fiber surface enhances the interfacial bonding between the fiber and the matrix, preventing interfacial separation and strengthening the mechanical properties of the material.
- The literature review is incomplete. A detailed review on fibers chemical treatment could be found in Thapliyal, D.; Verma, S.; Sen, P.; Kumar, R.; Thakur, A.; Tiwari, A.K.; Singh, D.; Verros, G.D.; Arya, R.K. Natural Fibers Composites: Origin, Importance, Consumption Pattern, and Challenges. J. Compos. Sci. 2023, 7, 506. 。
My answer according to this question: Thank you very much for your valuable suggestions. According to your suggestion, we improved the literature review section of the article and cited the references you recommended. Once again, thank you for the recommended articles, which have helped to make the literature review section of my article more standardized and comprehensive.
Detailed revisions can be found
The study conducted by Thapliyal et al.[13] revealed that the utilization of waste natural fibers in the production of fiber composites not only addresses environmental concerns but also offers a viable approach towards achieving circular economy objectives. Chaishome et al.[14] treated flax fibers with alkali and utilized them as fillers to prepare composite materials, resulting in enhanced thermal stability of the corresponding composites. Du et al.[15] conducted silane coupling treatment on pulp fibers, followed by melt blending with PLA, to fabricate silane-coupled pulp fiber-reinforced PLA composite materials. Their research revealed that the silane coupling-treated pulp fibers achieved reactive compatibility with PLA, resulting in improved mechanical properties compared to untreated counterparts. Goriparthi et al.[16] utilized 5% NaOH, potassium permanganate acetone, benzoyl peroxide, and silane solutions for surface treatment of jute fibers, followed by fabrication of jute fiber/PLA composite materials. Their investigation revealed that the surface-treated composites exhibited higher abrasion resistance compared to untreated counterparts. Among various chemical reagents, PLA/silane-treated jute composite materials demonstrated the optimal abrasion resistance, whereas PLA/alkali-treated jute composite materials exhibited the lowest abrasion resistance. Surface modification of fibers can indeed enhance various properties of composites. However, there is relatively limited research on the relationship between the chemical composition, surface functional group distribution of fibers after modification, and the various properties of composite materials. Additionally, there is scant investigation into the regularities governing the relationship between the internal component morphology of fibers and the respective mechanisms affecting material properties.
- Please, ellaborate more on moisture absorption which is a main effect of fibers chemical treatment.
My answer according to this question: Thank you very much for your valuable suggestions. According to your suggestion, we have added a discussion on the moisture absorption of the composite materials in the results and discussion section.
Detailed revisions can be found on page 17-18, lines 651-685:
3.10 Moisture Absorption Analysis
The moisture absorption of the composite material was evaluated by conducting water absorption tests, The comparison of moisture absorption for the composite materials is shown in Figure 11. In comparison, it can be observed that the SA-CSF component demonstrates superior moisture absorption characteristics. This is attributed to the presence of stearic acid, which forms a coating on the surface of cotton stalk fibers. Stearic acid, being a long-chain fatty acid, contains hydrophilic groups such as carboxyl groups in its molecular structure. These hydrophilic groups facilitate interactions with water molecules and attract them into the polylactic acid structure. Moreover, treatment with stearic acid effectively eliminates impurities protruding from cotton stalk fiber surfaces, enhancing their smoothness and providing an efficient pathway for water molecules to permeate into the material's interior, thereby further augmenting its moisture absorption capability. Alkali treatment removes components such as pectin, hemicellulose, and lignin from the surface of the cotton stalk fibers, resulting in increased surface roughness, irregular and porous features, which hinder the entry of water molecules into the material, thus weakening its moisture absorption capability. While silane treatment results in a cleaner surface of the cotton stalk fibers, the hydrophobic nature of the silane layer on the surface impedes the infiltration of water molecules, resulting in a decrease in the moisture absorption performance of the S-CSF component. The NS-CSF component has the worst moisture absorption performance among all composite materials. This is because the cotton stalk fibers in this component undergo alkali treatment, which removes pectin, hemicellulose, lignin, and other components from the fiber surface. As a result, the roughness of the fiber surface increases and disrupts the channels for water molecules to penetrate. Furthermore, silane treatment is applied to cover a hydrophobic silicon oxide coating on the surface of these fibers, further repelling water molecule infiltration and reducing their moisture absorption performance even more. The NS-CSF component exhibits the poorest moisture absorption performance among all composite materials due to the alkali treatment of cotton stalk fibers, which eliminates pectin, hemicellulose, lignin, and other components from the fiber surface. Consequently, the increased roughness of the fiber surface disrupts water molecule penetration channels. Additionally, a silane treatment is applied to create a hydrophobic silicon oxide coating on these fibers' surfaces, further deterring water infiltration and reducing their moisture absorption performance. The conclusion can be drawn that the enhancement of cotton stalk fiber surface polarity and improvement in overall fiber integrity and smoothness can significantly augment the moisture absorption performance of composite materials.
Figure.10 Moisture Absorption of Composites
References
[1] G. Rajeshkumar, S.A. Seshadri, G.L. Devnani, M.R. Sanjay, A.R. Anuf, Environment friendly, renewable and sustainable poly lactic acid (PLA) based natural fiber reinforced composites – A comprehensive review, Journal of Cleaner Production 310(1) (2021) 127483.
[2] M.H. Hussin, N.A. Pohan, Z.N. Garba, M.J. Kassim, A.A. Rahim, N. Brosse, M. Yemloul, M.R.N. Fazita, M.K.M. Haafiz, Physicochemical of microcrystalline cellulose from oil palm fronds as potential methylene blue adsorbents, International Journal of Biological Macromolecules 92 (2016) 11-19.
[3] Z. Lv, J. Xu, C. Li, L. Dai, H. Li, Y. Zhong, C. Si, pH-Responsive Lignin Hydrogel for Lignin Fractionation, ACS Sustainable Chemistry & Engineering 9(41) (2021) 13972-13978.
[4] G.P. Bernardes, M. de Prá Andrade, M. Poletto, Effect of alkaline treatment on the thermal stability, degradation kinetics, and thermodynamic parameters of pineapple crown fibres, Journal of Materials Research and Technology 23 (2023) 64-76.
[5] F.E. Bendourou, G. Suresh, M.A. Laadila, P. Kumar, T. Rouissi, G.S. Dhillon, K. Zied, S.K. Brar, R. Galvez, Feasibility of the use of different types of enzymatically treated cellulosic fibres for polylactic acid (PLA) recycling, Waste Management 121 (2021) 237-247.
[6] M.A. Sawpan, K.L. Pickering, A. Fernyhough, Effect of fibre treatments on interfacial shear strength of hemp fibre reinforced polylactide and unsaturated polyester composites, Composites Part A: Applied Science and Manufacturing 42(9) (2011) 1189-1196.
[7] C. Liao, Y. Xiao, K. Chen, P. Li, Y. Wu, X. Li, Y. Zuo, Synergistic modification of polylactic acid by oxidized straw fibers and degradable elastomers: A green composite with good strength and toughness, International Journal of Biological Macromolecules 221 (2022) 773-783.
[8] S. Kumar, R. Dang, A. Manna, N.K. Dhiman, S. Sharma, S.P. Dwivedi, A. Kumar, C. Li, E.M. Tag-Eldin, M. Abbas, Optimization of chemical treatment process parameters for enhancement of mechanical properties of Kenaf fiber-reinforced polylactic acid composites: A comparative study of mechanical, morphological and microstructural analysis, Journal of Materials Research and Technology 26 (2023) 8366-8387.
[9] Y. Long, Z. Zhang, K. Fu, Y. Li, Efficient plant fibre yarn pre-treatment for 3D printed continuous flax fibre/poly(lactic) acid composites, Composites Part B: Engineering 227 (2021) 109389.
[10] A. Mlhem, B. Abu-Jdayil, M.Z. Iqbal, High-performance, renewable thermal insulators based on silylated date palm fiber–reinforced poly(β-hydroxybutyrate) composites, Developments in the Built Environment 16 (2023) 100240.
[11] Properties of Poplar Fiber/PLA Composites: Comparison on the Effect of Maleic Anhydride and KH550 Modification of Poplar Fiber, International journal of applied mechanics 12(3) (2020).
[12] M.H. Hamidon, M.T.H. Sultan, A.H. Ariffin, A.U.M. Shah, Effects of fibre treatment on mechanical properties of kenaf fibre reinforced composites: a review, Journal of Materials Research and Technology 8(3) (2019) 3327-3337.
[13] D. Thapliyal, S. Verma, P. Sen, R. Kumar, A. Thakur, A.K. Tiwari, D. Singh, G.D. Verros, R.K. Arya, Natural Fibers Composites: Origin, Importance, Consumption Pattern, and Challenges, Journal of Composites Science 7(12) (2023) 506.
[14] L. Jichun, Influence of Surface Basification on Thermal Stability of SMA and SMA/OMMT, Modern Plastics Processing and Applications (2010).
[15] J. Du, Y. Hu, S. Hu, Interfacial control and dispersion characteristics of polylactic acid fiber/pulp fiber composites, Polymer Composites n/a(n/a).
[16] B.K. Goriparthi, K.N.S. Suman, N. Mohan Rao, Effect of fiber surface treatments on mechanical and abrasive wear performance of polylactide/jute composites, Composites Part A: Applied Science and Manufacturing 43(10) (2012) 1800-1808.

Round 2
Reviewer 1 Report
Comments and Suggestions for Authors
I consider that all corrections have been made.
Reviewer 2 Report
Comments and Suggestions for Authors
The authors revised the manuscript accordingly.
Reviewer 3 Report
Comments and Suggestions for Authors
This work is accepted.